# Transcriptomics analysis reveals molecular alterations underpinning spaceflight dermatology
Henry Cope [1,25], Jonas Elsborg [2,3,25], Samuel Demharter [3], J. Tyson McDonald [4], Chiara Wernecke [5,6], Hari Parthasarathy [5,7], Hriday Unadkat [5,8], Mira Chatrathi [5,9], Jennifer Claudio [5,10], Sigrid Reinsch [5,11], Pinar Avci [12], Sara R. Zwart [13], Scott M. Smith [14], Martina Heer [15], Masafumi Muratani [16,17], Cem Meydan [18], Eliah Overbey [18], Jangkeun Kim [18], Christopher R. Chin [18], Jiwoon Park [18,19], Jonathan C. Schisler [20], Christopher E. Mason [18,19], Nathaniel J. Szewczyk [1,21], Craig R. G. Willis [22], Amr Salam [23] & Afshin Beheshti [10,24] ✉

## Abstract

**Background** Spaceflight poses a unique set of challenges to humans and the hostile spaceflight environment can induce a wide range of increased health risks, including dermatological issues. The biology driving the frequency of skin issues in astronauts is currently not well understood.

**Methods** To address this issue, we used a systems biology approach utilizing NASA's Open Science Data Repository (OSDR) on space flown murine transcriptomic datasets focused on the skin, biochemical profiles of 50 NASA astronauts and human transcriptomic datasets generated from blood and hair samples of JAXA astronauts, as well as blood samples obtained from the NASA Twins Study, and skin and blood samples from the first civilian commercial mission, Inspiration4.

**Results** Key biological changes related to skin health, DNA damage & repair, and mitochondrial dysregulation are identified as potential drivers for skin health risks during spaceflight. Additionally, a machine learning model is utilized to determine gene pairings associated with spaceflight response in the skin. While we identified spaceflight-induced dysregulation, such as alterations in genes associated with skin barrier function and collagen formation, our results also highlight the remarkable ability for organisms to re-adapt back to Earth via post-flight re-tuning of gene expression.

**Conclusion** Our findings can guide future research on developing countermeasures for mitigating spaceflight-associated skin damage.

## Plain language summary

Spaceflight is a hostile environment which can lead to health problems in astronauts, including in the skin. It is not currently well understood why these skin problems occur. Here, we analyzed data from the skin of space flown mice and astronauts to try and identify possible explanations for these skin problems. It appears that changes in the activation of genes related to damage to DNA, skin barrier health, and mitochondria (the energy-producing parts of cells) may play a role in these skin problems. Further research will be needed to confirm exactly how these changes influence skin health, which could lead to solutions for preventing and managing such issues in astronauts.

Throughout the course of a spaceflight mission, the astronaut exposome includes altered gravity, elevated radiation, and confinement within a closed environment with unique hygiene procedures, light cycles, and ventilation[1]. These stressors perturb biological systems, inducing gene regulatory changes, mitochondrial dysregulation, microbiome shifts, and DNA damage[1]. Dermatological issues are not typically regarded as a key risk to astronaut health and mission success, yet they are amongst the most common in-flight health issues reported by astronauts. During regular International Space Station (ISS) missions averaging 6-months, skin rashes

have been identified as the most frequently reported in-flight clinical symptom, with 1.1 cases per flight year, accounting for 40% of all notable medical events and a 25-fold increase compared to the United States' general population[2]. An additional 0.3 cases per flight year include skin manifestations accompanied by symptoms of infection; skin lesions associated with viral reactivation were also reported and have been studied in-flight[2–4]. Notably, an in-flight skin rash was reported to have occurred during a 6-month ISS mission, worsening immediately after extravehicular activity (EVA)[5], and a post-flight skin rash was reported following the 1-year-long

NASA Twins spaceflight study[6]. These events arouse particular concern for future Moon and Mars missions, which will be longer, include high-levels of EVA, and also include risk of exposure to irritant dust, reported previously to induce skin issues in Apollo astronauts[7]. Elucidating the biological response of skin in space could aid development of new countermeasures to manage dermatological issues and optimize astronaut performance during these future missions.

As the human body's largest organ, the skin plays a crucial role in numerous essential health functions. These include facilitating fluid diffusion, promoting wound healing, regulating body temperature, and enabling tactile sensation. Skin is also the first line of defense against chemicals, allergens, infectious agents, and - importantly for spaceflight - radiation. Studies of the astronaut skin microbiome have identified microbial interchange between the skin and the ISS interior[8], and have also hypothesized that abnormal proliferation of certain types of opportunistic microorganisms on astronaut skin may stem from the unusual hygiene procedures on the ISS, where wipes are used as opposed to showering[9,10]. While these microbiome shifts and their associated health effects require further studies, investigations into the molecular response of skin tissue in space are lacking. In murine skin, the 13-day STS-135 study and the 91-day Mouse Drawer System (MDS) study reported significant spaceflight-induced modulation of extracellular matrix (ECM) genes[11,12]; the set of genes did not overlap, which could be due to significant differences in study design (e.g. duration). The MDS study also reported a 42% increase in synthesized procollagen coupled with a 15% reduction in dermal thickness likely indicative of increased collagen turnover, and an increase in hair follicles growing in the anagen stage accompanied by dysregulation of hair follicle genes[12]. Corroborating gene regulatory changes associated with hair cycle were also reported in an analysis of hair follicle samples from 10 astronauts in a Japan Aerospace Exploration Agency (JAXA) study[13], yet reports of skin physiological changes in astronauts, including dermal atrophy, have been mixed[14,15]. In a recent analysis of astronaut skin punch biopsies taken before and after the 3-day commercial Inspiration4 (i4) mission[16], gene expression changes within different layers of the skin were explored alongside microbiome changes, revealing signatures of increased inflammation and mitochondrial dysregulation across skin compartments, and gene regulatory changes associated with activation of DNA damage response (DDR) processes, T cell immunity, and increased barrier disruption in the outer epidermal layer of the skin[17].

In this study, we perform an analysis of five previously unreported murine skin RNA-Seq datasets from the JAXA Mouse Habitat Unit 2 (MHU-2)[18], NASA Rodent Research 5 (RR-5) and 7 (RR-7) experiments, to identify global gene expression signatures in spaceflight-exposed skin and to investigate the effect of study design on biological signatures relating to skin health and common spaceflight themes including DNA damage and repair, and mitochondrial dysregulation[1]. In addition to examining gene regulatory patterns, we use an explainable artificial intelligence (AI) modeling approach to construct interpretable machine learning models to identify synergistic effects between pairs of genes, which we interpret as possible biological interactions, that reveal putative disequilibria of dependent processes. We then investigate gene regulatory changes observed in the rodent skin data in astronaut data, including JAXA hair follicle data[13], skin data from the i4 mission, and blood gene expression data from the NASA Twins[19], JAXA Cell-Free Epigenome (CFE), and i4 studies. We conclude by suggesting how these molecular signatures may eventually lead to follow-up studies and pharmaceutical interventions.

## Methods
### Ethics and inclusion statement
This manuscript has included authors from all backgrounds from the scientific international community and the results are held at the highest ethical standards.

All human studies were done with ethical approvals with established and approved IRBs and according to the criteria set by the Declaration of Helsinki. The Inspiration4 (I4) mission was performed at Weill Cornell Medicine. Blood samples were provided by SpaceX Inspiration4 crew members after informed consent for research use. The procedure followed guidelines set by Health Insurance Portability and Accountability Act (HIPAA) and operated under Institutional Review Board (IRB) approved protocols and informed consent was obtained. Experiments were conducted in accordance with local regulations and with the approval of the IRB at the Weill Cornell Medicine (IRB #21-05023569). All crewmembers consented to data and sample sharing.

The NASA Twin Study subjects signed informed consent according to the declaration of Helsinki and 14 CFR Part 1230 for collection and use of sample materials and data in research protocols at NASA and the collaborating institutions. Study protocols were approved by the NASA Flight Institutional Review Board (protocol number Pro1245) and all participating institutions. Also the data is hosted at the NASA Life Sciences Data Archive (LSDA). Informed consent was provided by all participants in the NASA Twin Study. The JAXA astronaut data was obtained from NASA's GeneLab Platform and this study used only published aggregated quantification values.

No ethical approval for this manuscript was required to use the JAXA astronaut data since this was publicly available data on the GeneLab platform. In addition the data made available on GeneLab was only provided as anonymized aggregate data.

### GeneLab spaceflight murine datasets
Five RNA-Seq datasets (OSD-238, OSD-239, OSD-240, OSD-241, OSD-254) were downloaded from the NASA OSDR via the API, and full dataset descriptions can be found on the dataset pages. These datasets are derived from whole murine skin samples from the MHU-2, RR-5, and RR-7 spaceflight experiments. For MHU-2, singly-housed male C57BL/6J mice were 9 weeks of age when flown on the ISS for 30 days; they were euthanized less than 1 day after return to Earth. Dorsal and femoral skin samples were extracted from MHU-2 mice, and 6 replicates in spaceflight microgravity and the matching 6 replicates in the 1G ground control were split into 2 sets of 3, with half fed a JAXA chow diet and the other half fed JAXA chow with supplemental FOS[18]. For RR-5, 30 week old BALB/c mice in shared housing were flown on the ISS for 30 days; following return to Earth, mice were given 30 days to recover before euthanasia and extraction of dorsal and femoral skin tissue. Finally, for RR-7, 11 week old C57BL/6J mice and C3H/HeJ mice were flown on the ISS for either 25 or 75 days before being euthanized on-orbit.

### RNA-Seq analysis of rodent datasets
Raw counts data, derived via a previously reported pipeline[20], were downloaded from the NASA OSDR and loaded with tximport (v1.28.0). ERCC genes were filtered out, as were genes with zero counts in the minimum number of samples corresponding to a single condition. Differential Gene Expression (DGE) analysis using the Wald significance test was performed on spaceflight microgravity and ground control samples using the R package DESeq2[21] (v1.40.2). For each data subset, microgravity spaceflight samples were contrasted with ground control samples and cook's cut off and independent filtering were not used. Mapping of mouse ENSEMBL gene IDs to gene symbols and human orthologs was done using the R Package biomaRt (v2.56.1) and ENSEMBL 105. For human orthologs, one-to-many mapping was supported. All heatmaps were generated using the R package ComplexHeatmap (v2.16.0). All DGE statistics for the rodent data, including the list of cross-mission genes, can be found in Supplementary Data 1.

### Pathway analysis of rodent datasets
For Over Representation Analysis (ORA), the R package clusterProfiler (v4.8.3) was used on vectors of genes and bar plots were made via ggplot2 (v3.4.4). For Gene Set Enrichment Analysis (GSEA), the R package FGSEA (v1.26.0) was used. GSEA was performed on all DEGs for all data subsets, with rank vectors consisting of HGNC symbols and corresponding sorted t-scores in the data subset. MSigDB human collections H, C2, and C5 were used (v7.4) and set sizes >15 were permitted. For the mitochondrial pathway analysis, human pathways from MitoPathways (v3.0), with some additional

custom curated pathways (Supplementary Table 1), were used and set sizes >1 were permitted. All heatmaps were generated using the R package ComplexHeatmap (v2.16.0). All GSEA results, including mitochondrial results, can be found in Supplementary Data 2.

### RNA-Seq data analysis on Twin Study samples

Longitudinal samples were collected from a male astronaut aboard the ISS and his identical twin on Earth during a 340 day mission including 6 months preflight and 6 months postflight follow-up, for a total of 19 time points for the flight subject and 13 time points for the ground subject. Blood was collected using CPT vacutainers (BD Biosciences Cat # 362760) per manufacturer's recommendations. For full details of sample separation and processing see ref. [19]. Briefly, samples collected on ISS were either frozen in −80 °C after separation of mononuclear cells by centrifugation (referred to as CPT), or returned to Earth in 4 °C in a Soyuz capsule and sorted into CD4, CD8, CD19 populations and a lymphocyte depleted (LD) fraction. Samples collected on Earth were either frozen for mononuclear cells or processed when fresh for sorted cell populations. To correct for the effects of ambient temperature exposure on RNA (approximately 36 h including landing and repatriation) control samples were created by simulating similar conditions to those that may occur during the ambient return and were compared to fresh blood collections from the same individual. RNA extraction, library prep and sequencing were completed per[19] using both ribodepletion or polyA selection kits.

As previously reported[19], generated sequences were trimmed using Trim Galore! (v0.4.1) and quantified to genes using kallisto[22] on ENSEMBL transcripts. Differentially expressed genes were called using DESeq2[21] on each cell type separately by comparing preflight, inflight and postflight groups, controlling for the normal biological variance within the 24 months using the longitudinal data of the ground twin and using the simulated ambient control samples as another covariate for sorted cells[20,23]. Reads were mapped onto an updated reference genome (GRCh38)[24].

### JAXA Cell-Free Epigenome (CFE) Study RNA quantification data

Aggregated RNA differential expression data and study protocols were shared through the NASA OSDR with accession number: OSD-530[25]. Plasma cell-free RNA samples for RNA-seq analysis were derived from blood samples collected from 6 astronauts before, during, and after the spaceflight on the ISS. Mean expression values were obtained from normalized read counts of 6 astronauts for each time point. In total 2-3 pre-flight samples (L-168 ± 14 days, L-112 ± 14 days, L-56 ± 14 days), 4 in-flight samples (L + 5 ± 1 day, L + 30 ± 7 days, L + 60 ± 14 days, L + 120 ± 14d or R-8 (−14d/+0d day)), and 3-4 post-flight samples (R + 3 ± 1 day, R + 30 ± 7 days, R + 60 ± 14 days, R + 120 ± 14 days) were collected per astronaut. As provided by OSDR, data points for all subjects were pooled into one group per mission phase.

### JAXA astronaut hair follicle data

Gene expression data from 10 JAXA astronauts' hair follicles[13] was downloaded from the NASA OSDR (OSD-174). Raw data for 60 total samples was processed using LIMMA with R/bioconductor[26]. Briefly, duplicate sample single-color Agilent microarray data was background corrected, filtered for low expression probes, and quantile normalized. Differential gene expression was measured between pre-flight, in-flight, and post-flight data points using p-values adjusted for False Discovery Rates (FDR) with the Benjamini–Hochberg method. As previously described[13] hair follicle samples were collected from each astronaut on 6-month ISS missions at two pre-flight time points (firstly between L-180 and L-90, and secondly between L-60 and L-14), two in-flight time points (firstly between L + 20 and L + 37, and secondly between R-20 and R-7), and two post-flight time points (firstly between R + 2 and R + 7, and secondly between R + 30 and R + 90).

### Inspiration4 (i4) astronaut sample collection

Four civilians, two males and two females, spent three days in Low-Earth Orbit (LEO) at 585 km above Earth. The mission launched from NASA

Kennedy Space Center on September 15th, 2021 and splashed down in the Atlantic Ocean near Cape Canaveral on September 18th, 2021. All biological data derived from the Inspiration4 mission were collected pre- and post-flight, and will be accessible via the Space Omics and Medical Atlas (SOMA)[27]. For this study, only data from blood samples and skin biopsies were used. Pre-flight samples were collected at L-92, L-44, and L-3 days prior to launch to space. Upon return, post-flight samples were collected at R + 1, R + 45, and R + 82 days.

Blood samples were collected before (Pre-launch: L-92, L-44, and L-3) and after (Return; R + 1, R + 45, and R + 82) the spaceflight. Chromium Next GEM Single Cell 5′ v2, 10x Genomics was used to generate single cell data from isolated PBMCs. Subpopulations were annotated based on Azimuth human PBMC reference[28]. Reads were mapped onto a recent reference genome (GRCh38)[24]. We followed the analysis pipeline as previously reported [Ref. 29].

For skin spatial transcriptomics data, 4 mm diameter skin biopsies were obtained from all Inspiration4 crew members, once before flight and as soon as possible after return (L-44 and R + 1). As previously described in detail[17], these biopsies were flash frozen and processed with the NanoString GeoMx platform. Based on staining images using fluorescent antibodies, a total of 95 freeform regions of interest (ROIs) were profiled across outer epidermal (OE), inner epidermal (IE), outer dermal (OD) and vascular (VA) regions. The IE and OE regions represented stratum basale, and stratum spinosum with stratum granulosum, respectively. GeoMx WTA sequencing reads from NovaSeq6000 were compiled into FASTQ files corresponding to each ROI and converted to digital count conversion files using the NanoString GeoMx NGS DnD Pipeline. From the Q3 normalized count matrix that accounts for factors such as capture area, cellularity, and read quality, the DESeq2 method was used to perform DGE analysis.

Records of vitamin D supplement consumption were not available for the i4 crew.

All human studies were done with ethical approvals with established and approved IRBs at Weill Cornell Medicine. Blood samples were provided by SpaceX Inspiration4 crew members after consent for research use. The procedure followed guidelines set by Health Insurance Portability and Accountability Act (HIPAA) and operated under Institutional Review Board (IRB) approved protocols and informed consent was obtained. Experiments were conducted in accordance with local regulations and with the approval of the IRB at the Weill Cornell Medicine (IRB #21-05023569).

### Astronaut physiological data

Data are reported from three human subject experiments conducted on the International Space Station: Nutritional Status Assessment (2006-2012), Dietary Intake Can Predict and Protect Against Changes in Bone Metabolism During Space Flight and Recovery (Pro K) (2010–2015), and Biochemical Profile (2013–2018). All protocols were reviewed and approved by the NASA Institutional Review Board and all subjects provided written informed consent.

These missions were 4–6 months in length, and these studies included blood and urine collections before, during, and after flight, with analysis of an array of nutritional and biochemical markers. Blood and urine samples were collected 2 or 3 times before flight: approximately Launch minus (L-) 180 days and L-45 days. In some cases, a third blood sample was collected (typically along with the L-45 collection), and these tubes were centrifuged and frozen for aliquoting after flight batched with the samples collected inflight. Blood samples were collected inflight, at approximately Flight Day (FD) 15, 30, 60, 120, and 180. Postflight samples were collected in the first 24-h after landing (designated return + 0, R + 0) and again 30-d later (R + 30). The R + 0 samples were not necessarily fasting, given the time of day and nature of return from flight. Of the 59 crewmembers reported herein: 8 returned on the Space Shuttle, with blood collection 2–4 h after landing; 51 landed in Kazakhstan, with 7 of them returning to Star City, Russia, with blood collection 8–10 h after landing; 44 were transported directly back to the Johnson Space Center in Houston, with blood collection approximately 24-h after landing. Pre- and postflight collections included

two 24-h urine collections, and inflight collections included one 24-h urine collection. These collection techniques have been previously described[30].

We report here vitamins and metabolites, oxidative stress and damage markers, inflammatory markers and cytokines, liver enzymes and endocrine indices. These were analyzed using standard techniques as previously reported[31].

As of this writing, data were available for 59 crewmembers (47 male, 12 female). Age at launch was $47.0 \pm 5.6$ y, body mass at launch was $79.2 \pm 11.8$ kg (M: $83.3 \pm 9.3$; F: $63.0 \pm 4.5$). Body mass index was $25.5 \pm 2.9$ kg/m2 (M: $26.4 \pm 2.6$; F: $22.3 \pm 1.5$).

All available data are reported here, although the reported n for any given test or session varies for a number of reasons, including: not all experiments had all analytes included, mission length differences for some crewmembers, schedule or other issues occasionally precluded sample collection, and methods changes over time. Repeated measures analysis of variance was conducted to test for differences during and after flight compared to preflight, and comparisons among time points were made using a Bonferroni t-test. Multiple comparisons were accounted for, and only those tests with $p < 0.001$ are reported. The data was plotted using R package ggplot2 (v3.3.5).

## QLattice symbolic regression modeling

We used symbolic regression (QLattice v3.0.1[32]) to construct models involving combinations of synergistic genes which map from the gene expressions to spaceflight status. For these models, we only distinguish between mice that went to space and mice that did not. Conventional statistical methods typically allow for calculating the effect of a single gene at a time through metrics such as p-values and false discovery rates. In contrast, symbolic regression models can reveal combinations of genes and modules that best predict spaceflight status. These could be linear combinations involving two or more genes that have previously also been shown to be statistically significant, or it could be non-linear combinations that reveal features that on their own were non-significant but in synergy with a second feature become highly predictive. In addition, known biological functions of genes included in models, as well as the resultant mathematical relationship between them, can potentially be interpreted to reveal regulations or interactions that are affected by spaceflight.

In biological pathway analysis, it is well-known that up- or down-regulation of one gene can have cascading effects, such that the function of one gene becomes sensitive to that of another[33]. It has previously been demonstrated that parsimonious machine learning models are able to provide accurate outcome prediction in omics data, while preserving interpretability[34,35]. The interpretability often results from the fact that models might demonstrate otherwise opaque relations, which become clear when combined effects are considered. An example could be an interaction where the regulation of a single gene is itself unimportant for functional changes, *unless* another gene is simultaneously regulated. This would indicate a synergistic compound effect, which naturally can expand to a 3-, 4- or 5-gene synergy and beyond. The combined effect of two or more individually insignificant gene expression levels may thus theoretically be more powerful than the effect of a single gene with low p-value. Thus, traditional statistical analysis masks such effects by the use of metrics related to the individual gene.

## UMAP dimensional reduction and gene clustering

For the clustering of genes shown in Fig. 1b, we performed a uniqueness filtering by first dropping all genes where more than a quarter of the samples had identical expression levels. Subsequently, we filtered out all genes with a variance of less than 1.7, resulting in 2184 filtered genes with high variability.

A dimensional reduction was performed subsequent to filtering by uniformly distributing the filtered data on a Riemannian manifold, using the Uniform Manifold Approximation and Projection for Dimension Reduction (UMAP). UMAP is a general purpose manifold learning and dimension reduction algorithm which is similar to t-SNE in that it predominantly preserves local structure. Yet, UMAP preserves more global structure, which makes it a more suited algorithm when the objective of the dimensional reduction is more than simple visualization (in this case the objective is clustering).

Genes were then clustered using the Hierarchical Density-Based Spatial Clustering of Applications with Noise (HDBSCAN)[36]. A primary advantage of HDBSCAN is that it is always deterministic for the same hyperparameters, and will thus always return the same clustering, all else being equal. In addition, comparable clustering algorithms such as k-means do not perform well unless clusters are of equal size and density with few outliers. With biological data such as gene expressions, we expect large variation in cluster size and density, making HDBSCAN the ideal choice.

Once genes were clustered, the gene sets belonging to each cluster were extracted and analyzed using three ontology databases in the Python implementation, GSEAPY, of the Gene Set Enrichment Analysis[37] tool Enrichr[38]. We used the Elsevier Pathway Collection, the 2021 WikiPathway Collection, and the 2021 MGI Mammalian Phenotype Level 4 Collection. These enrichment analyses were used to provide context to each cluster by appending an annotation if a notable amount of hits showed up for a particular association.

## Regulatory network analysis

We used the differential gene expression data from all cross-mission genes across each comparison to perform an upstream regulator analysis using QIAGEN Ingenuity Pathway Analysis (IPA) (version 01-22-01[39]). We identified the biological or chemical drugs in the QIAGEN IPA library of regulators, defined as having a Benjamini–Hochberg adjusted $P$ value $< 0.05$ and an absolute Z-score $>1$. Drugs from the dataset that met these criteria and were associated with similar $\log_2$ fold-change patterns in the QIAGEN Knowledge Base were visualized by hierarchical clustering of the unstandardized Z-scores using Euclidean distance as the similarity metric, and complete-linkage method for agglomeration. Statistics on the full list of drug targets can be found in Supplementary Data 3.

## Reporting summary

Further information on research design is available in the Nature Portfolio Reporting Summary linked to this article.

## Results

### Spaceflight transcriptome global changes in rodent skin reflect common biological hallmarks of spaceflight

To explore whether transcriptomic changes in the skin would occur during spaceflight, we analyzed five RNA-Seq datasets from the NASA Open Science Data Repository (OSDR)[40], all representing skin tissue from space flown mice and matching ground control replicates. These datasets are derived from three different spaceflight missions (Fig. 1a). The most variable genes across the datasets cluster into functional groups related to established physiological risks of spaceflight (Fig. 1b). For example, spaceflight is known to induce immune dysfunction[1] and Cluster 1 involves highly-correlated genes associated with immune response, including genes linked to modulation of Immunoglobulin G (IgG) levels. Microgravity is also well established to cause degradation of muscle[1], and Cluster 9 contains a significant number of genes related to muscular morphology and muscle disorders. Ionizing radiation is known to cause renal and pulmonary fibrosis[41,42] and profibrosis-like alterations were previously observed in lung tissue of mice returned from Space Shuttle Endeavour (STS-118)[43]. Renal and lung fibrosis associated genes found in Cluster 7, and genes encoding proteins involved in sclerotic diseases grouped in Cluster 8, further supports these reports.

Upon subsetting each dataset based on experimental groups (i.e. diet, strain, skin site) and performing differential gene expression (DGE) analysis for spaceflight samples versus ground control samples, we end up with 10 data subsets (Fig. 2a). In total 476 unique differentially expressed genes (DEGs) are significant (False Discovery Rate (FDR) $\leq 0.1$) in at least 2 out of the 10 data subsets. One significant DEG shared across data subsets from the

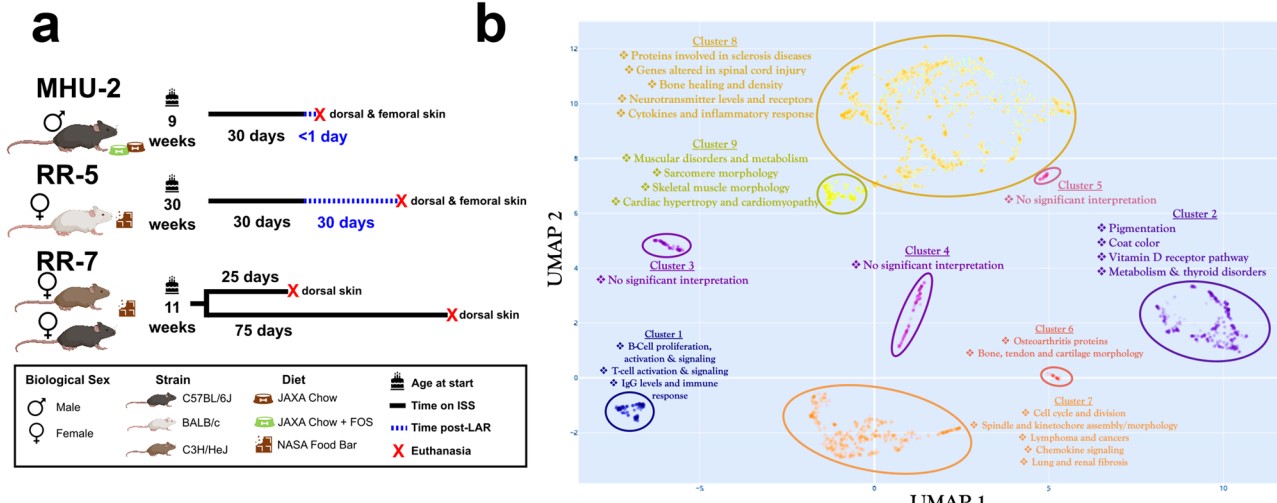

**Fig. 1 | Global data overview. a** Breakdown of the rodent datasets used in this study. **b** Clustering of the most variable genes within the rodent datasets with functional annotation.

RR-7 study is *GLYCAM1* which is shared between both data subsets for the C57BL/6 J strain. *GLYCAM1*, a ligand for L-selectin that has been proposed to modulate leukocyte transendothelial migration from blood[44], is strongly upregulated (Log$_2$ Fold Change (LFC) ≈ 18.81, FDR < 10$^{-6}$) at the 25 day time point and strongly downregulated (LFC ≈ −15.66, FDR < 0.002) at the 75 day time point, which could indicate a relationship between spaceflight duration and immune alterations for the C57BL/6J mice. Over representation analysis (ORA) reveals that a large proportion of the 244 significant DEGs shared between MHU-2 data subsets are associated with the organization of collagen and the ECM (Fig. 2b). Significant modulation of genes associated with ECM homeostasis was previously reported in analyses of the skin of space flown mice[11,12]. The MHU-2 subset from the dorsal skin of mice fed a standard JAXA chow diet with supplemental prebiotic fructooligosaccharides (FOS) contains the highest quantity of significant (FDR ≤ 0.1) DEGs by a substantial margin (Fig. 2a). In ground-based rodent studies, FOS has been shown to improve gut microbiome balance, increase bone density, and affect the immune system through increased short chain fatty acid (SCFA) production (modifying interleukin production and natural killer cell activity), and modification of the immune system via gut-associated lymphoid tissue[18,45]. Due to the disparity between the number of significant DEGs in the data subsets, and the variety of conditions between data subsets, care must be taken when combining these data subsets to infer shared responses of rodent skin to spaceflight.

**Genes significantly altered in rodent skin spaceflight response across multiple missions are associated with cell cycle processes**

To address the previously mentioned bias in the MHU-2 dorsal skin data subset with a FOS-supplemented diet, and to delineate the common response of murine skin to spaceflight across multiple datasets, a list of 189 Differentially Expressed Genes (DEGs) is generated exhibiting significance (False Discovery Rate ≤0.1) in data subsets from at least two distinct missions. For the majority of the significant DEGs, there is a trend of downregulation in the MHU-2 dorsal skin dataset with the FOS-supplemented diet, and in the RR-7 mission C57BL/6J mice after 75 days of flight (Supplementary Fig. 1). A smaller cluster of genes (e.g., *LAMA1, S1PR1, HMGCS2, TXNIP, IGFBP3, ADGRF5, CAR4, HSD17B11, C1QTNF9, KLF11*) are also significant in those same data subsets, but are significantly upregulated (Fig. 3a).

To go beyond single-gene significance, we use QLattice modeling[32] to construct models that predict the spaceflight status of all samples in the data (i.e. from all data subsets). These models use the expression of two genes at most to arrive at maximally relevant and minimally redundant models. In Fig. 3c, d, we show the decision boundaries of the two strongest models found, with Area Under Curve (AUC) scores of 0.86 and 0.85, respectively. Both models show that the spaceflight status is well separated using only the expression of two genes as input, while also having plausible biological interpretations, which we discuss in the following sections.

The 189 cross-mission DEGs are primarily involved in cell-cycle pathways (Fig. 3b), with several of the genes (e.g. *LAMA1, C1QTNF9, HMGCS2, TXNIP, KLF11*) clustering with average trends of upregulation across the datasets (Fig. 3a). Spaceflight is well-documented to perturb the cell-cycle[1]. These genes tend also to be involved in metabolic pathways, particularly those associated with diabetes; *HMGCS2* has been shown to regulate mitochondrial fatty acid oxidation[46], while *LAMA1* and *KLF11* variations have been shown to be risk factors in type 2 diabetes[47,48]. Aside from being a protector against oxidative stress, *TXNIP* is similarly implicated in metabolic diseases, and is typically upregulated in diabetic and prediabetic muscle tissue[49]. *C1QTNF9* (also known as *CTRP9*) has also emerged as a potentially important component of pathways involving lipid metabolism and adipose tissue, exemplified by the fact that *C1QTNF9* transgenic mice have been shown to resist weight gain and metabolic dysfunction[50]. It is interesting to note that upregulation of these metabolic pathway genes appears generally weaker for female mice. This might be linked to the same mechanisms offering higher protection for females against diet-induced obesity compared to male mice, as has been noted in other studies[51].

Sphingosine-1-phosphate (S1P)-Sphingosine-1-phosphate receptor (*S1PR*) axis have been implicated in pathogenesis of both inflammatory and autoimmune skin diseases and also plays a role in skin's immune response to viral infection[52]. In both MHU-2 and RR-7 missions *S1PR1* is enriched, while this effect is reversed in RR-5 after 30-day recovery (Fig. 3a). On the contrary *FLG* (Filaggrin), *CASP14, KRT2*, genes involved in cornified envelope and skin barrier formation[36–38,53–55], generally showed an opposite trend in gene expression patterns (Fig. 3a). *CASP14* is also the main driver of the second-most predictive 2-gene QLattice model obtained from the cross-mission key gene list, which involved a combined model of *CASP14* and *S1PR1* (Fig. 3d).

*OXTR*, identified as a key driver in the most predictive 2-gene QLattice model derived from the cross-mission key gene list, along with *SLC6A18* (Fig. 3c), is predominantly localized in the basal layer of the epidermis. Its primary expression is attributed to keratinocytes and dermal fibroblasts[56]. *OXTR* together with oxytocin has been shown to mediate oxidative stress response in the skin[56]. Our analysis revealed an increase in gene expression of *OXTR* across all missions and experimental conditions except for C3H/Hej mice in the RR-7 mission at the 25-day time point (Fig. 3a). Both

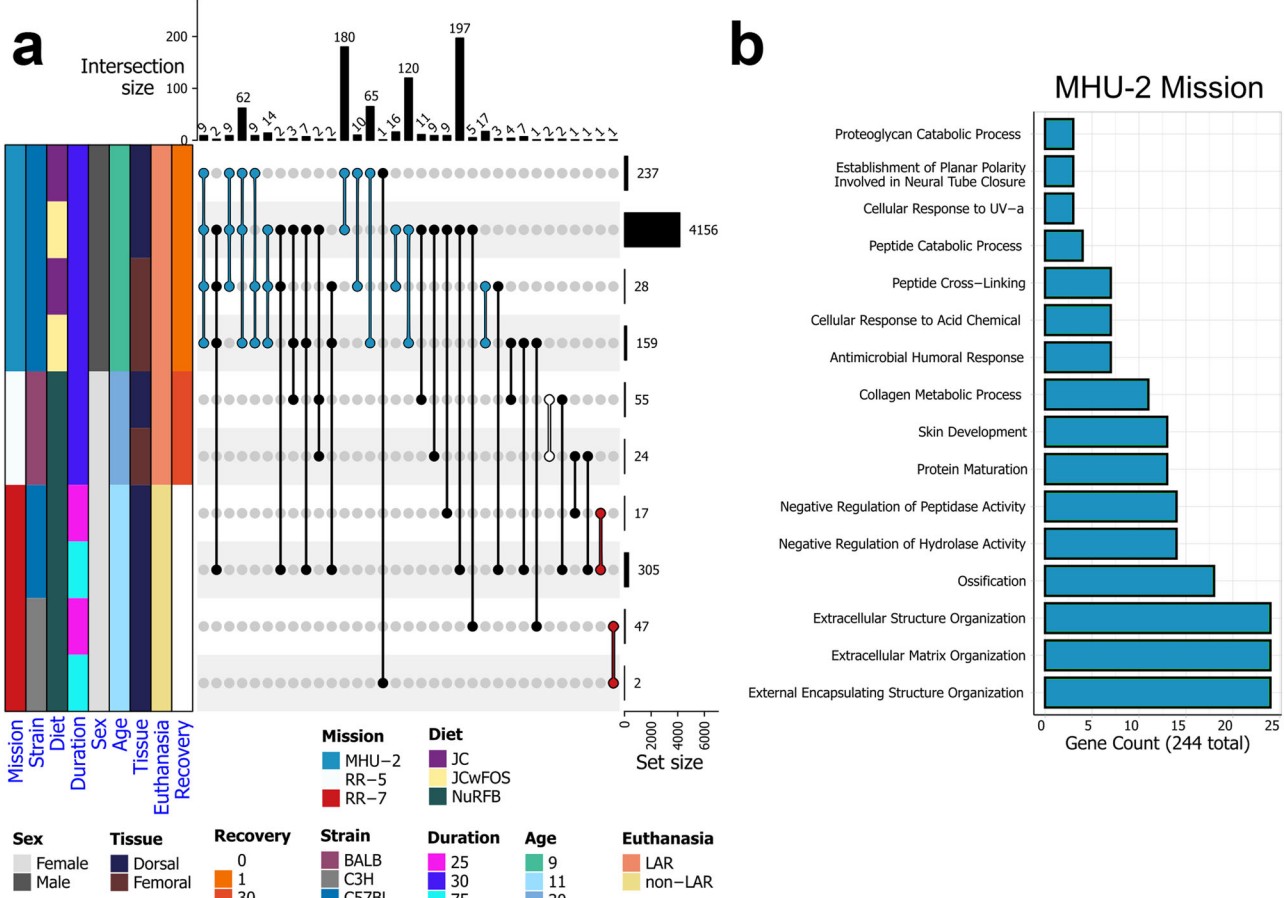

**Fig. 2 | Differentially expressed genes shared between datasets. a** An upset plot showing the number of significant (FDR ≤ 0.1) differentially expressed genes (DEGs) in spaceflight versus ground data subsets and the number of overlapping significant DEGs between these data subsets. The colored annotation bar on the left of the plot shows how the original datasets divide into 10 data subsets based on all unique conditions including diet, biological sex and strain. The bar plot on the right of the upset plot shows the number of significant DEGs in each of the 10 data subsets. The bar plot on the top shows the number of intersecting DEGs between combinations of the data subsets, as indicated by the connected dots within the body of the upset plot. Black connecting lines indicate combinations spanning across multiple missions, and other connecting lines are colored according to the annotation bar, based on their shared mission. **b** A bar plot of significant Gene Ontology Biological Process (GOBP) pathways from significant DEGs shared by any subsets within the MHU-2 mission (i.e., DEGs from the blue bars in panel a).

estrogens and androgens play an important role in skin and hair physiology. *HSD17B11*, which has a stronger expression in steroidogenic cells such as *sebaceous* glands, has been predicted to be involved in androgen metabolism during steroidogenesis, and conversion 5α-androstane-3α, 17β-diol to androsterone[57]. Similar to the pattern observed in *OXTR*, *HSD17B11* expression was increased in all murine datasets except for RR-5 dorsal skin group (Fig. 3a).

**Astronaut data correlates with select gene expression changes occurring across missions in the murine skin models**

Having investigated a set of cross-mission genes regulated by spaceflight in rodent skin, we then decided to investigate the expression of human orthologs of these genes in astronaut studies. Firstly, we use post-flight vs pre-flight gene expression data from skin samples collected during the i4 mission. Similar to the patterns observed in murine models (Fig. 3a), *FLG* and *CASP14*, which play significant roles in skin barrier formation, are also downregulated in the epidermis of i4 astronauts (Fig. 4a), where they are primarily expressed. In the literature, *FLG* is reported to mainly be expressed in stratum granulosum[58], with mutations associated with dry skin, atopic dermatitis, contact dermatitis as well as ichthyosis vulgaris[54,59,60]. *CASP14* on the other hand is known to be primarily expressed in differentiating and cornifying layers[44] and decreased levels were detected in lesional skin biopsies of patients with atopic dermatitis and contact dermatitis[61,62]. Likewise, *KRT2* which was downregulated across all subgroups in the

MHU-2 and RR-7 missions (Fig. 3a), was also downregulated in the skin of i4 astronauts, with strongest downregulation in outer epidermis where it is primarily expressed but also in inner epidermis and outer dermis (Fig. 4a). Additionally, similarly to the trend of upregulation in the rodent data (Fig. 3a), *OXTR* is upregulated across all compartments studied in the i4 skin (Fig. 4a) and *HSD17B11* is upregulated in both compartments of the epidermis and in the outer dermis (Fig. 4a).

In some astronauts returning from long-term spaceflight, decrease in melanin content has been previously reported[63]. In the inner epidermis of the i4 skin data, where melanin synthesis takes place, we found downregulation of two genes associated with melanin biosynthesis, *TYRP1* and *DCT*[64] (Fig. 4a). However, *TYR* which synthesizes tyrosinase, the rate limiting enzyme of melanin synthesis was upregulated in the same compartment[65] (Fig. 4a). *TYR, DCT*, and *TYRP1* clustered in the rodent data, with an average trend of downregulation, particularly in the MHU-2 dorsal skin dataset with the FOS-supplemented diet, and in the RR-7 mission C57BL/6J mice after 75 days of flight (Fig. 3a).

We also analyze gene expression data from the blood of astronauts from the NASA Twins, JAXA CFE studies, and i4 studies (Supplementary Fig. S2). In the NASA Twins Study, the most noteworthy results are in CD4 cell type in-flight vs pre-flight samples with significant downregulation in a group of genes (e.g. *DSC2, TMEM252, CEBPD, CA4*, and *PTGS2*, and *OTX1*) persisting post-flight (Supplementary Fig. 2a). These genes display no significant changes in CD8 cells, which may indicate that

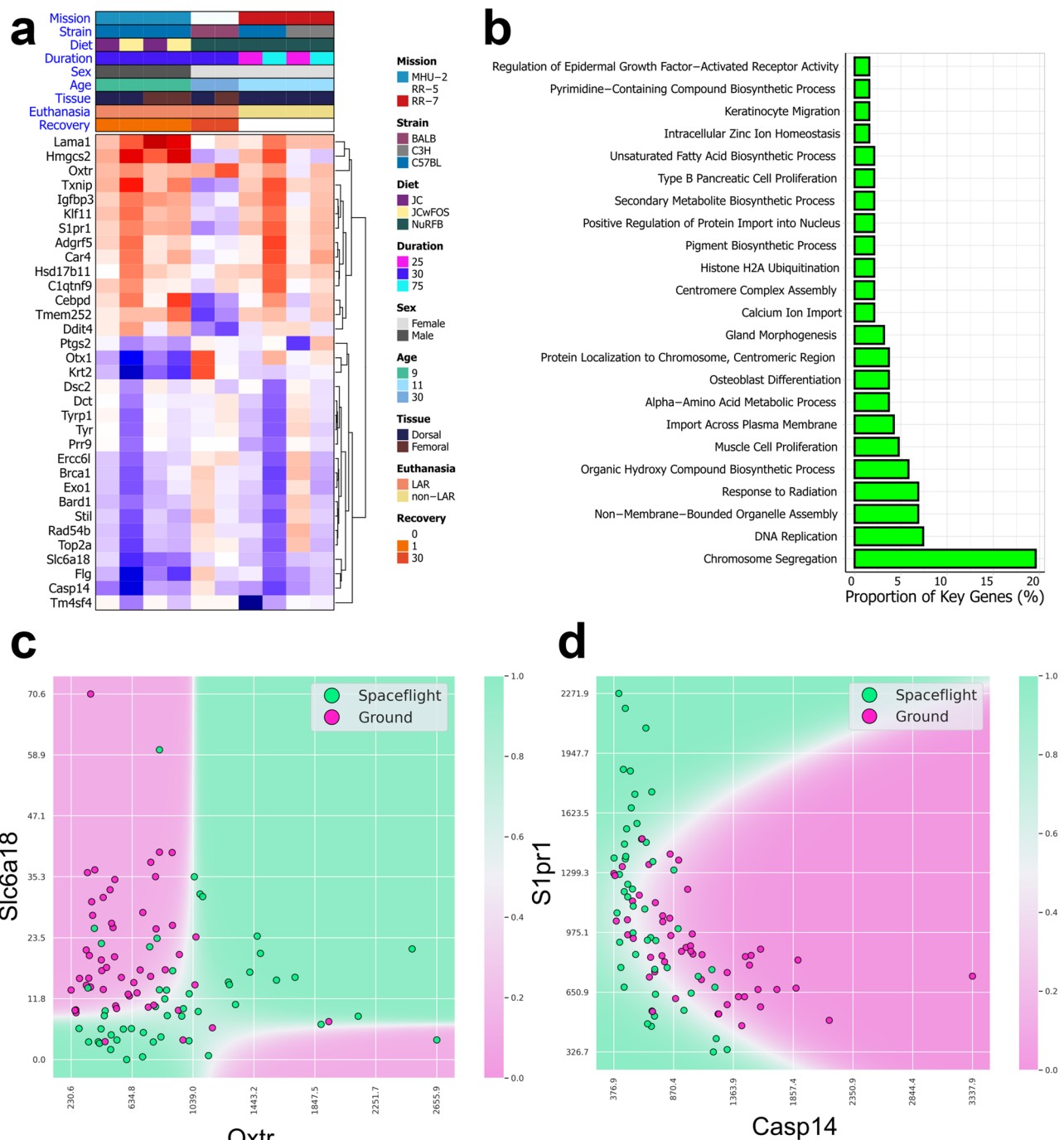

**Fig. 3 | Cross-mission genes involved in rodent skin spaceflight response. a** A heatmap showing regulatory changes of select cross-mission genes (genes that are significantly (FDR ≤ 0.1) differentially expressed between flight and ground control across multiple missions) within each rodent data subset. **b** A bar plot showing the GOBP pathways that the full list of cross-mission genes are involved in. **c** A decision boundary plot of the strongest cross-mission QLattice model using only the expression of two genes as input features to predict the spaceflight status of the all mice in the data. **d** A decision boundary plot of the second-strongest model across all missions.

their downregulation is a response to immune system stressors affecting the helper function and signaling of CD4 cells. *DDIT4* (DNA-damage-inducible transcript 4), *STIL* and *TOP2A* cell-cycle associated genes are amongst the most strongly upregulated genes in-flight compared to pre-flight in the CD8 cells, and exhibit downregulation post-flight to trend towards pre-flight levels (Supplementary Fig. 2a). In the CFE study, several genes were strongly upregulated in-flight and then downregulated post-flight (e.g., *OXTR, PRR9, ERCC6L*) in the blood samples from JAXA astronauts (Supplementary Fig. 2b). In the i4 mission blood samples the

majority of genes show a pronounced suppression in the time point immediately before flight, which could be indicative of increased stress prior to launch. Several genes then show strong upregulation at the first post-flight time point (e.g., *BARD1, BRCA1, RAD54B, EXO1, OXTR,* and *HSD17B11)* (Supplementary Fig. 2c), suggesting upregulation due to spaceflight or landing stressors, followed by post-flight recovery towards baseline expression levels from second post-flight time point and beyond. However, some other genes appear to show a delayed upregulation following spaceflight, as they are upregulated in the second post-flight time

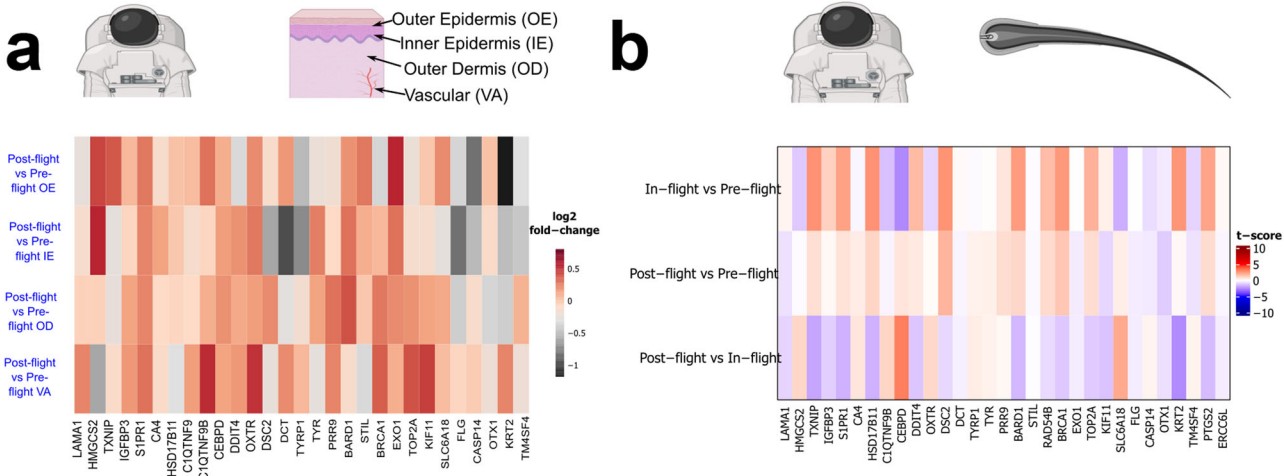

**Fig. 4 | The profile of the cross-mission genes in astronauts. a** Heatmap showing log₂ fold-change in orthologs of the rodent skin cross-mission genes in astronaut skin data from the Inspiration4 mission, for different skin layers. **b** Heatmap showing t-score in orthologs of the rodent skin cross-mission genes in hair follicle samples at different time points from the JAXA HAIR astronaut study.

point, two months after flight (e.g. *KLF11, CA4, OXT1, TM4SF4*) and return towards baseline at the third post-flight time point (Supplementary Fig. 2c). The gene expression alterations in these sets of genes are potentially due to a re-adaptation response.

### Specific pathways and genes related to skin health are altered in spaceflight

To determine direct relevance of spaceflight to skin health, we conduct targeted analysis with a curated list of key pathways involved in skin health from the Molecular Signatures Database (MSigDB) and identify which of these pathways and associated genes are significantly modulated in the murine skin datasets. A collagen biosynthesis pathway and a set of collagen genes (e.g., *COL1A1, COL1A2, COL3A1, and COL5A1*) were significantly (FDR ≤ 0.1) suppressed/downregulated across all dataset subsets for the MHU-2 mission, while showing a trend of weak enrichment/upregulation across the two other missions (Fig. 5a). Additionally, several collagen and/or ECM related genes (e.g., *MMP3, COL5A2, COL6A1, COL6A2, COL15A1, COL6A3, SPARC, PCOLCE, and PYCR1*) were significantly (FDR ≤ 0.1) downregulated in all of the MHU-2 data subsets except femoral skin with the standard diet, and a few collagen related genes were significantly downregulated only in the dorsal subsets from the MHU-2 mission (e.g., *P3H1, CRTAP, COL14A1, and FKBP14*) (Fig. 5a). Overall, the distinct collagen/ECM downregulation signature (Fig. 5a) suggests that a study design difference in the MHU-2 mission (compared to the RR-5 and RR-7 missions), where the use of young single-housed male mice or dissection shortly after the hypergravity event of Live Animal Return (LAR) may have suppressed collagen biosynthesis. Similarly, we find an overall trend of enrichment of pathways relating to thin skin and dermal atrophy in the RR-5 and RR-7 mice, with suppression occurring in the MHU-2 mice (Fig. 5a), likely driven by regulatory patterns in the collagen-associated genes. While there is a general trend of upregulation, the RR-5 mission generally lacks any significant results for genes relating to skin health, which could be due to the 30 days of recovery post-flight. In addition to being reported as cross-mission genes in the previous section, *CASP14, FLG* and *KRT2* were also identified as key skin health genes (Fig. 5).

In post-flight vs pre-flight skin samples from the i4 mission, several collagen/ECM genes (e.g., SPARC, *COL1A1, COL1A2, COL3A1, COL6A1, COL6A2,* and *COL6A3*) were downregulated across the whole dermis, whereas *COL5A1* was only downregulated in the vascular compartment of the dermis (Fig. 5b). *MMP3*, which induces accelerated loss of skin collagen when there is inflammation, was upregulated across the whole dermal layer[65] (Fig. 5b). Likewise, several genes (e.g., *P3H1, COL14A1, COL15A1,* and *COLA12A1*) were upregulated in both compartments of the dermis (Fig. 5b).

*KRT24*, involved in terminal differentiation of keratinocytes, is reported to be mainly confined to stratum spinosum with slight overexpression in senescent keratinocytes[49] and has been shown to induce apoptosis of keratinocytes[66]. In the post-flight vs pre-flight i4 skin data *KRT24* is upregulated in all compartments of the skin with a more prominent increase in outer epidermis and vascular layers[66] (Fig. 5b). Late cornified envelope genes (LCE) are located in the epidermal differentiation complex on chromosome 1 and in the skin their expression is known to be mainly confined to the epidermis, specifically to upper stratum granulosum and premature cornified envelopes[67,68]. In both the inner epidermis and outer dermis of the i4 skin data, *LCE2A, LCE2B, LCE2C, LCE2D*, genes that respond to UV and calcium[67] and *LCE5A* were upregulated post-flight vs pre-flight (Fig. 5b).

In the NASA Twins study blood samples, *COL6A2* was upregulated in-flight and then downregulated post-flight in CD8 cells, with *PYCR1* following the same pattern in CD4 cells (Supplementary Fig. 3a). In the CFE study, blood samples from JAXA astronauts had strong in-flight upregulation of *RPTN* and *COL3A1*, and in particular *LCE2B*, where upregulation increased further post-flight (Supplementary Fig. 3b). On the other hand, *PYCR1* and *GJB6* showed the opposite trend in gene expression, with strong downregulation in-flight and almost complete reversal upon return (Supplementary Fig. 3b). *PYCR1*, which was upregulated post-flight vs pre-flight in all skin layers of i4 astronauts (Fig. 5b) encodes pyrroline-5-carboxylate reductase 1, an enzyme that catalyzes the last step in proline synthesis via utilization of *NADH* or *NADPH*. Proline plays an important role in collagen synthesis and *PYCR1* deficiency results in collagen and elastin abnormalities[69]. Conversely, *COL3A1* encodes alpha 1 chain of type III collagen, which is an EMP synthesized in cells as pre-collagen and its accumulation was shown to cause fibrosis in various organs[70,71]. In case of *LCE2C* and *KRT33A*, weak in-flight downregulation was followed by strong post-flight upregulation (Supplementary Fig. 3b). Although less apparent, this trend for *KRT33A* was also observed in the JAXA HAIR data (Fig. 5c). *KRT33A*, one of the type 1 hair keratin genes, is known to be highly expressed in the cortex of hair follicles, and its expression is reported to decrease with age[72]. Lastly, genes such as *COL6A2* and *FKBP14*, a collagen/ECM related gene associated with diseases such as kyphoscoliotic Ehlers-Danlos syndrome that is characterized by joint hypermobility, hyperelastic skin, hearing impairment, muscle hypotonia and occasional vascular fragility like aortic rupture[73], were downregulated in-flight and their average expression persisted post-flight (Supplementary Fig. 3b).

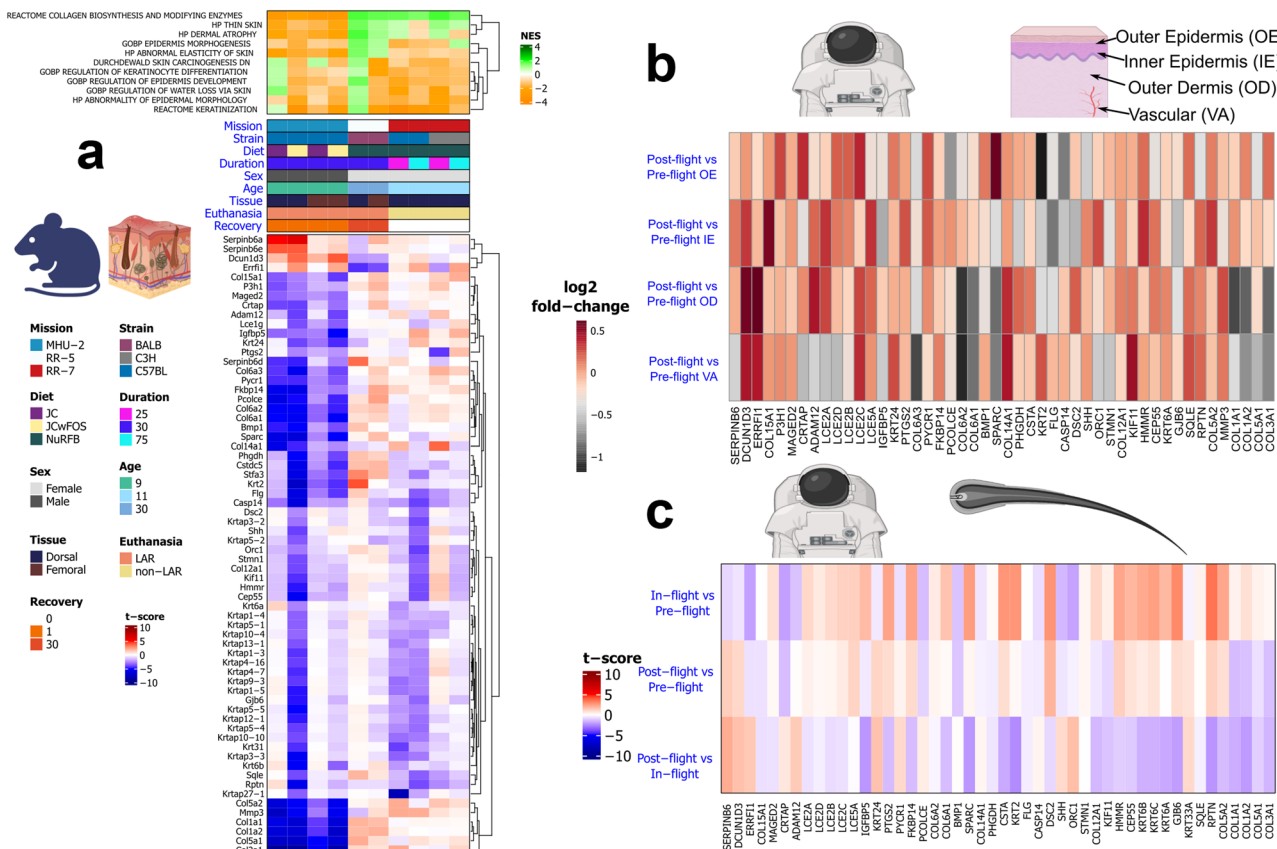

**Fig. 5 | Behavior of the specific genes associated with skin health in astronauts and rodents. a** The orange (suppressed) and green (enriched) heatmap shows the normalized enrichment score (NES) of curated skin health pathways (full list Supplementary Table 1) that are significant (FDR ≤ 0.05) in at least one rodent subset. The red (upregulated) and blue (downregulated) heatmap shows the t-score for leading edge genes from the significant pathways that are significant (FDR ≤ 0.1) in at least two data subsets. **b** Heatmap showing log₂ fold-change in orthologs of the genes from Panel **a** in astronaut skin data from the Inspiration4 mission, for different skin layers. **c** Heatmap showing t-score in orthologs of the genes from Panel **a** in hair follicle samples at different time points from the JAXA HAIR astronaut study.

## Radiosensitivity of mouse strains impacts DNA damage response in the skin following spaceflight

As part of our targeted analysis, we also opted to investigate the modulation of DNA damage and repair pathways in rodent skin. DNA damage and repair is a well-established response to space radiation[19,74] and little research is reported on the consequences this will have on the skin during spaceflight. When ionizing radiation hits DNA molecules, single-strand breaks (SSBs) and double-strand breaks (DSBs) occur, with DDR mechanisms activated to repair these breaks[74]. For the majority of DNA damage and repair pathways, both the dorsal and femoral skin RR-5 mission data subsets and the 25-day time point from the C3H/HeJ mice in the RR-7 mission show an opposing pattern compared to the other data subsets (Fig. 6). In these three data subsets, pathways relating to DNA damage and repair are generally enriched, while being suppressed in other data subsets. BALB/c and C3H/HeJ mice, as used in these data subsets, have been shown to be more radiosensitive compared to C57BL/6J mice[75–77], so repair mechanisms may be activated to mitigate increased radiation-induced DNA damage. In the C3H/HeJ mice there is a trend of decreasing activity for the DDR pathways from the 25-day to 75-day time points which could indicate either adaptation of the DNA repair pathways over time in space or dysregulation following extended-duration spaceflight (Fig. 6). The enrichment of DDR pathways in RR-5 also follows 30-days of spaceflight and a post-flight recovery period of 30-days (Fig. 6). However, DNA repair genes were reported to still be enriched compared to pre-flight levels when evaluated at 6-months post-flight during the NASA Twins study[19], so persistent DDR activation during post-flight recovery in the mice could be indicative of a similar mechanism.

## Mitochondrial dysregulation increased in the skin during spaceflight

Mitochondrial stress has been identified as a key hub for spaceflight response in multi-tissue analysis, yet skin was not included[78]. Skin is a tissue with high turnover and energy requirements[79], so we decided to investigate spaceflight changes relating to mitochondrial stress in murine skin. We find that spaceflight alters mitochondrial pathways in the skin (Fig. 7a), as previously observed in other tissues and across species[78]. Enrichment of an integrated stress response (ISR) pathway in all but two of the dataset subsets (Fig. 7a) is consistent with a previous report of potentially activated ISR due to mitochondrial dysfunction in space flown mice[78]. Interestingly, the two data subsets where ISR appeared to instead be suppressed were the 25 day time points for both strains from the RR-7 mission (Fig. 7a), implicating spaceflight duration as an important factor for modulating ISR. Additionally, we ran the QLattice machine learning model on the full gene set to obtain a set of models that were unbiased by feature selection due to e.g. variance filtering and the selection of cross-mission genes. One of the 10 strongest 2-feature models involved the expression of the genes *D2HGDH* (D-2-hydroxyglutarate dehydrogenase)[80] and *RPLP0-PS1* (a gene coding for the ribosomal protein RPLP0). The model's decision boundary (Fig. 7b) demonstrates how the simultaneous upregulation of *D2HGDH* and downregulation of *RPLP0-PS1* characterizes the space-flown mice, with a resulting AUC score of 0.82[80]. The upregulation of *D2HGDH* generally indicates an increased ability to break down the toxic D-2-hydroxyglutarate (D2H) compound in the mitochondria. In our case, this could be a compensatory mechanism due to larger build-up of the compound, and the model suggests that this is further dependent on the abundance of the RPLP0 protein.

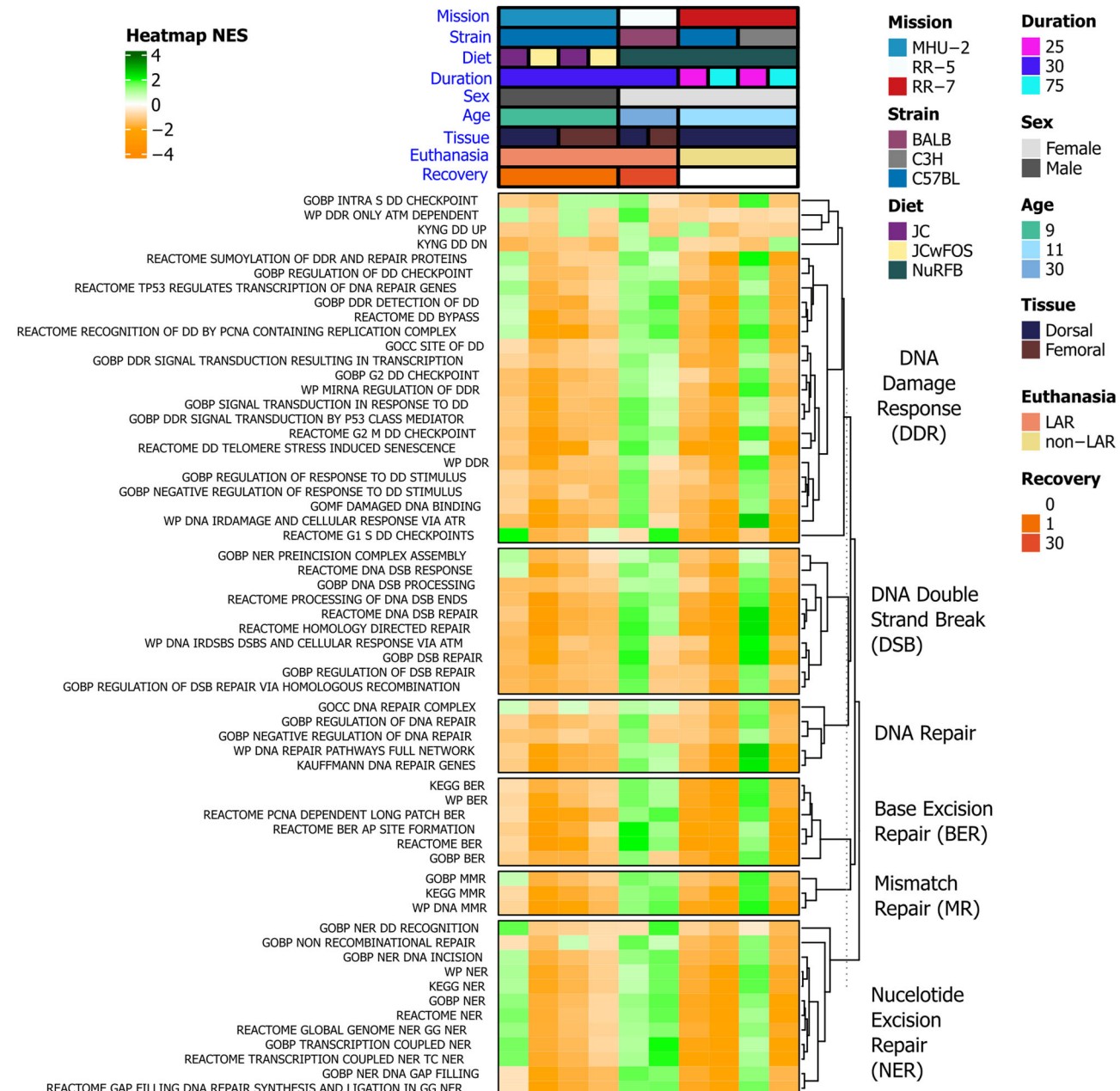

**Fig. 6 | DNA damage and repair pathways being regulated in rodents flown to space.** Heatmap of pathways relating to DNA damage response and repair mechanisms (full list Supplementary Table 1), significant (FDR ≤ 0.05) in at least one data subset.

## Circulating physiological markers from astronauts indicate that exercise countermeasures may have improved skin health

We investigate standard physiological biomarkers collected from astronaut urine and blood[81,82] to see whether average trends in these biomarkers connect to spaceflight gene regulatory changes occurring in the skin (Fig. 8). In-flight increases in IGF-1, leptin and white blood cell levels may indicate altered stress response due to spaceflight. Following a hypothesis that improved countermeasures may improve skin physiological parameters in astronauts on the ISS[14,15], we split the data into astronauts that used the older Interim Resistive Exercise Device (iRED), and astronauts that used the newer Advanced Resistive Exercise Device (ARED). Notably, overlapping data to that shown in Fig. 8 have been used as evidence of improvements in parameters relating to bone mineral density[81]. Indeed, while vitamin D decreased in-flight and normalized upon return to Earth for both exercise devices, ARED appeared to reduce this drop. Vitamin D is well-established to be an important mediator of skin health[83]. Notably, this decrease in

vitamin D is observed despite consistent programmatic vitamin D supplementation since the collection of blood samples[81].

## Spaceflight-induced gene regulatory changes in the skin correlate with relevant drug responses

Having established a list of genes that changed in the rodent skin across multiple missions, we then investigate potential drug targets and mechanisms for these genes via Ingenuity Pathway Analysis (IPA) (Fig. 9). As with the DDR pathways (Fig. 6), both the dorsal and femoral skin RR-5 mission data subsets and the 25-day time point from the C3H/HeJ mice in the RR-7 mission show an opposing pattern compared to the other data subsets for the predicted upstream regulators (Fig. 9).

Several drugs were predicted to exhibit significant positive correlation with gene expression changes seen in the space flown mice, with the exception of dorsal skin tissues obtained from C3H/HeJ mice that spent 25 days in flight for the RR-7 mission, and dorsal and femoral skin from the

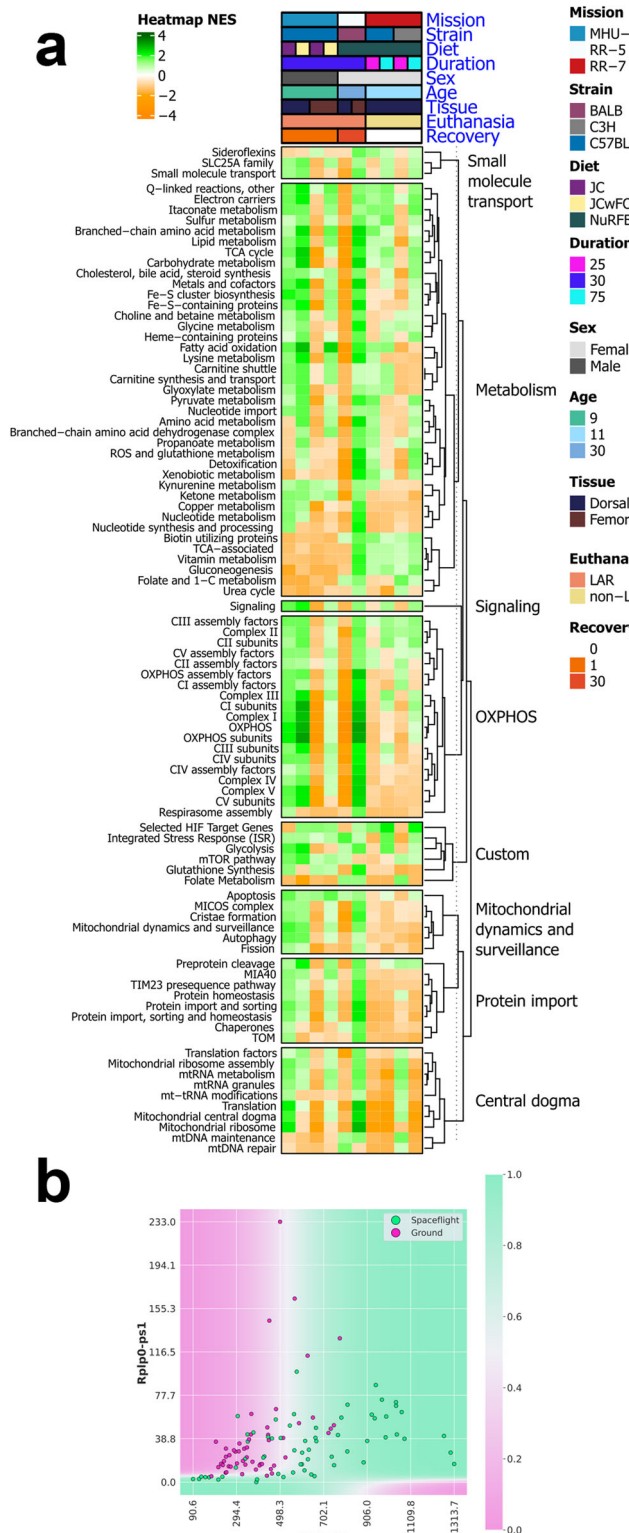

**Fig. 7 | Mitochondrial specific analysis on rodent spaceflight skin tissue.**
**a** Heatmap of pathways relating to the mitochondria, significant (FDR ≤ 0.05) in at least 1 data subset. **b** Decision boundary for the 2-gene model related to mitochondrial changes. The model indicates an increased removal of the toxic D2H compound in mitochondria through upregulation of *D2HGDH*, which is less pronounced when *PPP1R3B* expression is suppressed.

RR-5 mice (30-day recovery) (Fig. 9). For example, calcitriol is the active form of Vitamin D, and has been observed to affect skin barrier formation and epidermal differentiation[84] it is used clinically to treat plaque psoriasis[85]. Calcitriol is also associated with immune system alterations in the skin, and topical application in mice has been shown to exacerbate atopic dermatitis[86]. On the other hand, in a murine model, epidermal permeability and anti-microbial skin barrier impaired by corticosteroid, was shown to be reduced via calcitriol[87]. Alteration of gene regulatory patterns associated with calcitriol, in the mouse data subsets (Fig. 9) could indicate activation of compensatory mechanisms for spaceflight-induced skin barrier dysfunction, epidermal differentiation, and immune system alterations. L-asparaginase showed the strongest correlation with gene regulatory changes in these spaceflight-exposed mice (Fig. 9). L-asparaginase is used in treatment of acute lymphoblastic leukemia and lymphoblastic lymphoma and exerts its action by depriving leukemic cells of circulating asparagine, resulting in DNA breaks, cell cycle arrest and cell death[87,88]. It is also an immunogenic protein and leads to skin rash, dry skin, and hypersensitivity reactions[89]. Similarly, there was strong positive correlation between gene regulatory changes in these spaceflight-exposed mice associated with use of the Vitamin A derivative tretinoin (all-trans retinoic acid)[90], and two selective estrogen receptor modulators, tamoxifen and 4-hydroxytamoxifen[91–94].

On the contrary, estrogen and diethylstilbestrol, a synthetic form of estrogen, showed a reversed pattern compared to the aforementioned drug targets (Fig. 9). These drugs showed positive correlation with the expression changes in the RR-5 datasets, and the RR-7 25-day C3H/Hej data subset, and negative correlation in the other data subsets (Fig. 9).

## Discussion

Skin is well-established as an essential organ for health on Earth, yet despite the frequency of dermatological issues in astronauts[2], the molecular response of skin to spaceflight is understudied. Here, we have performed a comprehensive study on the molecular impact of spaceflight on skin, with the aim of addressing gaps in the understanding of spaceflight associated skin health risks. Our analysis indicated intriguing gene regulatory changes occurring in the skin of spaceflight vs ground control mice during the MHU-2, RR-5, and RR-7 spaceflight experiments, that also show some overlaps with molecular alterations observed in astronauts.

The genes exhibiting the highest variability across the datasets formed clusters associated with well-established physiological risks of spaceflight, notably immune dysfunction and muscle degradation (Fig. 1b). Furthermore, the majority of genes showing significant differential expression across multiple missions in murine data were linked to cell cycle processes, particularly chromosome segregation (Fig. 3a, b). This suggests that skin mirrors common spaceflight signatures observed in other tissues, including dysregulation of cell cycle processes[1,19]. Further to that point, we see an enrichment of DDR pathways in some data subsets from radiosensitive BALB/c and C3H/HeJ strains[75–77] (Fig. 6), which is also a well-established spaceflight response[19,74]. These repair mechanisms may be activated to mitigate radiation-induced DNA damage and appear to remain active during the post-flight recovery process, as seen in the RR-5 mission where samples were collected 30 days after flight. This finding is consistent with reports of sustained post-flight DDR activation from the NASA Twins study[19]. Notably, DNA damage in the skin has important implications for skin cancers including melanoma, of which astronauts carry a higher risk, although the direct correlation between spaceflight and melanoma is still under investigation[95,96]. Similarly, mitochondrial dysregulation is established as a key molecular signature of spaceflight[78], and we see enrichment of an ISR pathway in all but two of the murine skin data subsets (Fig. 7a), which may be driven by increased oxidative stress or DNA damage. ISR activation appears to be impacted by spaceflight duration, as the 25 day time points for RR-7 show a suppression of the ISR pathway[1,78]. Using an explainable machine learning approach[32], we also observed the counteractive mechanism

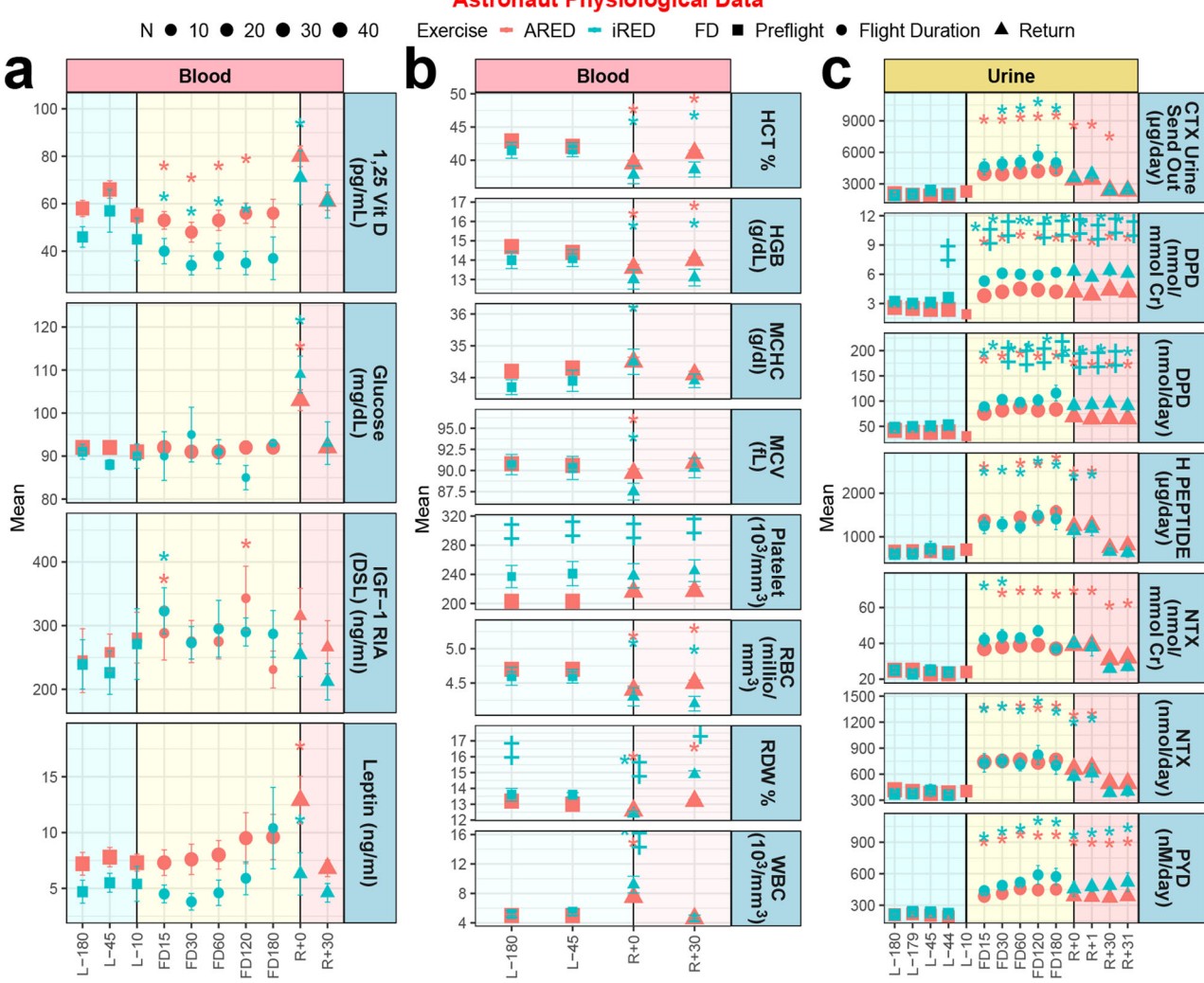

**Fig. 8 | Astronaut physiological markers compiled from up to 50 astronauts.**
**a** Specific blood markers which contain data points for pre-launch (L−), flight (FD), and return to Earth (R+). The numbers on the x-axis indicate the number of days for each group. Interim Resistive Exercise Device (iRED) is shown in blue and Advanced Resistive Exercise Device (ARED) is shown in red. **b** Specific blood markers which contain data points for pre-launch (L−) and return to Earth (R+). **c** Specific urine markers which contain data points for pre-launch (L−), flight (FD), and return to Earth (R+). The statistics on the data are *$p < 0.001$ for significantly different from L-45 and ‡$p < 0.01$ significantly different from ARED.

played by *D2HGDH*, which was generally upregulated in space flown mice (Fig. 7b) indicating an increased need for breakdown of toxic D-2-hydroxyglutarate (D2H) molecules in mitochondria. In this model, the partner gene *RPLP0-PS1* codes for the ribosomal protein RPLP0 and is down-regulated in space flown mice. RPLP0 has previously been implicated in the top significant network for the mitochondria-mediated cycle of Alzheimer's disease[97]. Low or absent levels of RPLP0 seem to limit the compensatory mechanism in the model (Fig. 7b). If this occurs, build-up of D2H and suppression of certain enzyme functions can result, causing DNA and histones to enter hypermethylated states and activate oncogenes and inhibit tumor suppressors[98]. Build-up of D2H in the mitochondria may thus be related to cell changes induced by DNA damage from space radiation. Notably, gene regulatory changes associated with mitochondrial dysregulation and DDR were also reported in the skin data from the i4 mission[17], corroborating these changes as key molecular hallmarks of spaceflight in the skin.

The downregulation of collagen/ECM genes and the suppression of collagen/ECM-related pathways in the MHU-2 mouse mission, where spaceflight samples were collected shortly after landing (Figs. 2b and 5a),

mirror findings from previous studies where a reduction in collagen synthesis occurred under hypergravity conditions in cultured human fibroblasts[99]. In contrast, RR-7 mice (frozen in space) generally exhibited an upregulation of the same collagen/ECM genes (Fig. 5a), consistent with results from the aforementioned human fibroblast study, which reported a 143% increase in collagen synthesis during microgravity[99]. Similar to RR-7, RR-5 also showed a trend of upregulation in collagen/ECM genes (Fig. 5a). The opposing patterns between the MHU-2 mission and RR-5/RR-7 may arise from diverse factors owing to experimental design differences. Notably, the timing of sample collection introduces a critical variable, as seen in the MHU-2 mission where samples were collected less than one day post-flight, in contrast to RR-5, where collection occurred after a 30-day post-flight recovery, and RR-7, where samples were obtained directly in space. This temporal distinction implies that collagen suppression observed in MHU-2 mice may be influenced, at least in part, by the acute hypergravity response, a known factor in decreasing collagen synthesis[99], and the stress-induced release of glucocorticoids[100–102], rather than being solely attributed to the spaceflight environment. Moreover, differences in sex and housing arrangements further complicate the comparison; MHU-2 mice are singly-

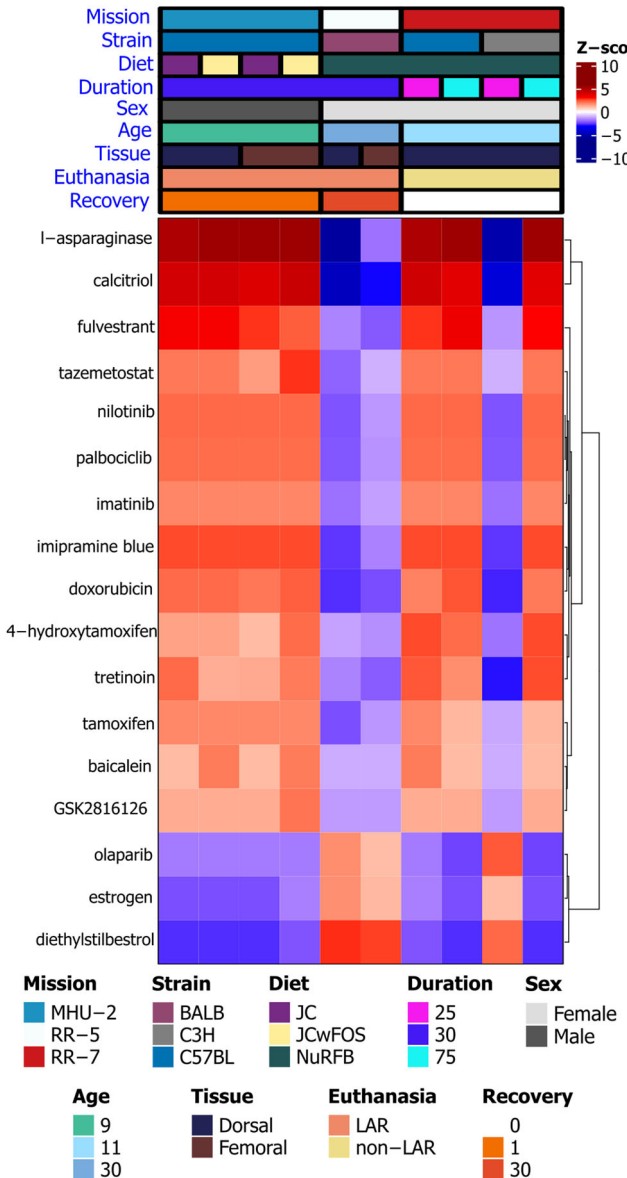

**Fig. 9 | Predicted drug signatures associated with spaceflight response in the skin.** Predicted drug signatures using the cross-mission genes across each dataset represented by a hierarchically clustered heat map. A positive (red) activation state implies cross-mission gene expression changes are consistent with expression changes observed with the indicated drug from curated causal gene expression relationship studies. A negative (blue) activation state implies the cross-mission gene expression changes are opposite to changes observed with the indicated drug.

housed males, while RR-5 and RR-7 mice are multi-housed females, introducing additional variables that may contribute to the observed disparities[100-103].

Skin has an anatomically heterogeneous gene expression profile[104]. For instance, fibroblasts exhibit distinct gene expression patterns based on their positional identity in relation to the major anatomical axis[105]. This phenomenon in part explains the predisposition of different sites to different skin conditions[104]. Although several genes related to collagen synthesis and ECM were downregulated across all four MHU-2 experimental groups, some of these genes showed statistical significance in the dorsal group but not in the femoral group (Fig. 5a). Likewise in the RR-5 mission, during terrestrial recovery, skin biopsies obtained from dorsal skin generally showed a higher reversal of gene expression pattern compared to those obtained from the femoral skin (Figs. 3a and 5a). These changes may be indicative of the variations in gene expression profile, their adaptation

response to space flight and re-adaptation to terrestrial environment, at different anatomical sites.

Skin biopsies collected from i4 astronauts one day post-flight exhibited a marked downregulation of various collagen/ECM-associated genes in the dermal layer compared to pre-flight conditions (e.g., *SPARC, COL1A1, COL3A1, COL6A1*) (Fig. 5b). However, certain genes, such as *SPARC* and *COL3A1*, showed an upregulation in one or both compartments of the epidermis (Fig. 5b). Moreover, several other collagen/ECM associated genes (e.g. *COL14A1, COL15A1, MMP3*) were upregulated in both epidermis and the dermis (Fig. 5b). As with the MHU-2 mice, the i4 astronauts' skin biopsies were obtained one-day post-flight, so some of these effects may have been induced by hypergravity. Nevertheless, blood data obtained from the same i4 astronauts at 1-, 45-, and 82-days post-flight suggests that most gene alterations observed upon landing revert back to normal levels within 82 days following flight, so we may expect a similar response in the skin tissue (Supplementary Figs. 2c and 3c). Previously reported findings from the NASA Twins showed in-flight activation of collagen formation pathways and an increase in COL1A1 and COL3A1 proteins detected in the urine, indicative of structural remodeling[19]. These changes, as with the majority of spaceflight-induced changes, were also reported to revert to baseline post-flight[19].

Some of the changes reported by astronauts during and after spaceflight, such as dryness and contact dermatitis, point to an impaired barrier function[17,95] our results support this hypothesis. *FLG* is a gene expressed in the stratum granulosum layer of the epidermis, during the epidermal differentiation process[41]. It encodes profilaggrin, a major component of keratohyalin granules found in the keratinocytes that is dephosphorylated and cleaved into filaggrin[106,107]. During conversion of keratinocyte cell membrane to cornified cell envelope, filaggrin aggregates and aligns intermediate filaments such as keratin 1 and keratin 10, within the cytoskeleton of keratinocytes, and subsequently a cross-linking process is initiated by proteins such as involucrin and locicrin[108,109]. This results in compactization and flattening of the corneocytes, one of the constituents of the skin barrier[110]. We identified *FLG* as one of the cross-mission genes in the spaceflight vs ground control mice data with downregulation across the majority of data subsets (all except RR-5 dorsal skin) (Fig. 3a). Additionally, we observed post-flight vs pre-flight downregulation of *FLG* in the epidermis of i4 astronauts (Fig. 5b). Caspase-14, an enzyme encoded by the *CASP14* gene, cleaves filaggrin, breaking it down into free amino acids. These amino acids serve as integral components of natural moisturizing factors, contributing to the preservation of skin hydration and pH[111]. Additionally, they play a crucial role in antimicrobial defense, further contributing to the multi-faceted functions of the skin barrier[61,112,113]. Similar to *FLG, CASP14* demonstrated downregulation in the majority of murine skin subsets (excluding RR-5 dorsal skin) (Fig. 3a), the epidermal layer of i4 astronaut skin tissue (Fig. 5b), and JAXA hair follicles (Fig. 5c). This observation is important considering the crucial role *CASP14* plays in skin barrier formation and its involvement in skin conditions such as contact dermatitis and atopic dermatitis[53,61,62,114]. The link between filaggrin deficiency and dry skin, altered pH, atopic and contact dermatitis, and ichthyosis vulgaris is also well-known[54,60,112,115-117].

*CASP14* was also the driver of one of the two main QLattice models we found to be maximally relevant and minimally redundant among the cross-mission genes, resulting in a very simple model involving only *CASP14* and *S1PR1* with an AUC of 0.85 (Fig. 3d). *S1PR1* was upregulated in most murine skin subsets (Fig. 3a), the in-flight vs pre-flight samples from the JAXA hair follicle study (Fig. 4b), and in all compartments for the post-flight vs pre-flight i4 astronauts' skin samples (Fig. 4a). In the RR-5 mission, after a 30-day recovery, in-flight effects may have been reversed as *S1PR1* expression was downregulated (Fig. 3a), which is also consistent with downregulation in the post-flight vs pre-flight samples from the JAXA hair follicle study (Fig. 4b). Downregulation of *S1PR1* has been reported to be necessary for activation of resident memory CD8 + T cells in skin[35] and has previously been observed in CD8 + T cells from murine skin during response to Herpes Simplex Virus Infection[118]. While the S1P-S1PR axis has

inhibitory roles in keratinocyte proliferation, it was also shown to contribute to the process of chemotaxis[119]. For instance, in Staphylococcus-aureus-stimulated keratinocytes; TNF alpha, IL-8 and IL-36γ release was induced via involvement of *S1PR1* and *S1PR2*[120]. Higher expression levels of *S1PR1* and *S1PR2* observed in the skin of impetigo patients and changes in the distribution patterns of these receptors were also aligned with these findings[120]. Herpes Simplex Virus reactivation[3,4,95] and Staphylococcus colonization in the skin[10,17,121] (which could also be influenced by breakdown of filaggrin altering pH levels[122]) have been widely reported in astronauts, therefore alterations in *S1PR1*, *CASP14*, and *FLG*, are of importance.

At present, there is a lack of widely accepted countermeasures to effectively mitigate the damage caused to the skin induced by exposure to the space environment. Our analysis has identified drugs capable of altering similar molecular changes that manifest in the skin during spaceflight (Fig. 9). Several drugs, including calcitriol, tretinoin, and tamoxifen showed significant positive correlation with regulatory changes in all data subsets from the MHU-2 and RR-7 missions except for the C3H/HeJ mice samples collected at the 25-day time point (Fig. 9). Tretinoin, a vitamin A derivative is a retinoid known for its applications in treating acne and leukemia, exerts its effects through the activation of retinoic acid receptors, influencing various skin processes such as increased proliferation and inhibited differentiation of epidermal cells, increased keratinocyte turnover, increased collagen synthesis, inhibited UV radiation induced metalloproteinase activity, and melanin reduction[23,123,124]. Tretinoin treatment on reconstructed human epidermis was shown to induce loss of keratohyalin granules and reduce filaggrin and Keratin 10 expression[125]. Retinoids have also been observed to downregulate caspase-14 expression in in-vitro reconstructed skin models[126]. Dry skin and exfoliation are common side effects during its use[123]. Tamoxifen, a selective estrogen receptor modulator, has been shown to exhibit both agoniztic and antagonistic effects depending on the tissue[94]. Its use in breast cancer patients is linked to cutaneous adverse events, including dry skin, itching, and occasionally radiation recall dermatitis and baboon syndrome (a Type IV hypersensitivity reaction)[126–129]. Furthermore, tamoxifen induces androgen receptor expression in hair follicles, leading to tamoxifen-induced alopecia in some cases[130], and abnormal hair follicles, epidermal atrophy and dermal fibrosis have been observed in rat models[131]. On the other hand, only olaparib, estrogen, and diethylstilbestrol (a synthetic form of estrogen) showed an opposing pattern, with significant negative correlation in all data subsets except for those from the RR-5 mission and the C3H/HeJ mice samples collected at the 25-day time point of the RR-7 mission (Fig. 9). Estrogen plays a crucial role in skin health, influencing keratinocyte function, collagen production, moisture retention and telogen-anagen follicle transition[132–135]. In an explant model of fetal rat skin development, estrogen was shown to increase filaggrin expression[136]. Conversely, estrogen deficiency during menopause is associated with adverse skin changes such as loss of collagen, skin dryness and atrophy[137,138]. Overall, molecular alterations observed in the space flown mice and astronaut skin data show similarities with molecular effects associated with drugs like tretinoin and tamoxifen[123], such as *FLG* and *CASP14* downregulation, while symptoms reported in astronauts including dry skin and dermatitis also resemble side effects associated with drugs like tretinoin and tamoxifen[106] (Fig. 9). Topical moisturizers and emollients, as commonly used by astronauts[15,95], may help to counteract the physiological consequences of disrupted barrier function genes, and may also help to explain why results on impact of spaceflight on skin physiological parameters in astronauts have been mixed[14,15], despite physiological evidence of skin damage in space flown rodents[12].

In conclusion, we have provided a comprehensive study on the impact of spaceflight on skin health. Currently, there is a gap in knowledge for how space radiation and microgravity affects skin biology, yet many of the common themes of spaceflight dysfunction emerged in our analysis, suggesting that skin could be an easily-accessible candidate for studying the biological impact of spaceflight. Our systems biology analysis revealed some genes of interest and potential mechanisms to investigate in the context of countermeasures that can be utilized in future studies. We believe with our study we have started to address the current gaps and provided some clues on how to potentially mitigate the adverse effects of the space environment to the skin.

The rodent datasets used come from three different spaceflight experiments with a variety of confounding variables associated with differences in study design. This means that direct comparison between the experiments is challenging, but indeed this also presents an opportunity to hypothesize how these factors may influence biology, as done in this manuscript. A lack of physiological data, such as dermal thickness, and lack of individual skin layer data from the rodent datasets means that generated hypotheses relating to mouse physiology cannot be confirmed without follow-up investigations, but the comparison to astronaut data helps translate the results to human relevance. Lastly, gene expression levels differ between skin compartments so an identical fold-change value in a layer where the specific gene expression is generally low may not represent the same magnitude in a layer where the same gene's expression is relatively greater. As such, for the i4 skin data we focused on changes in the compartments where the gene is established to be predominantly expressed.

## Data availability

For all the sequenced data underlying the plots in this manuscript the datasets used are publicly available via the NASA OSDR's Biological Data Management Environment (https://osdr.nasa.gov/bio/repo), including murine skin RNA-Seq datasets (OSD-238, OSD-239, OSD-240, OSD-241, OSD-254), microarray data from the JAXA hair study (OSD-174), and the JAXA CFE data (OSD-530). To create the UMAP plot in Fig. 1b, the normalized count data from all datasets was used. Genes where more than 25% of samples had identical expression levels were dropped, and from the remaining 75% of genes, we furthermore dropped genes with a variance of less than 1.7. The normalized count data used for the UMAP plot can be found at NASA Open Science Data Repository (OSDR), at the following links: OSD-238, OSD-239, OSD-240, OSD-241, and OSD-254. For each dataset, scroll down and click on Files -> GeneLab Processed RNA-Seq Files -> Normalized counts data. The source data for the rest of the figures are available in Supplementary Data 4. Deposited data from the sequencing data from the NASA Twin Study can be found on the NASA Life Sciences Data Archive (LSDA) and the accession code is not available due to privacy concerns. The astronaut physiological data is also available through LSDA due to privacy concerns. LSDA is the repository for all human and animal research data, including that associated with this study. LSDA has a public facing portal where data requests can be initiated[139]. The LSDA team provides the appropriate processes, tools, and secure infrastructure for archival of experimental data and dissemination while complying with applicable rules, regulations, policies, and procedures governing the management and archival of sensitive data and information. The LSDA team enables data and information dissemination to the public or to authorized personnel either by providing public access to information or via an approved request process for information and data from the LSDA in accordance with NASA Human Research Program and JSC Institutional Review Board direction. The Inspiration4 data has been uploaded to two data repositories: the NASA Open Science Data Repository (osdr.nasa.gov; comprised of NASA GeneLab and the NASA Ames Life Sciences Data Archive [ALSDA]), and the TrialX database. Identifiers for publicly downloadable datasets in the OSDR are documented as follows: 1) Data can be visualized online through the SOMA Browser (https://soma.weill.cornell.edu/apps/I4_Multiome/), the single-cell browser (https://soma.weill.cornell.edu/apps/I4_Multiome/), and the microbiome browser (https://soma.weill.cornell.edu/apps/I4_Microbiome/) and 2) for the PBMC data the data is available with OSDR accession ID: OSD-570 and the following link: https://osdr.nasa.gov/bio/repo/data/studies/OSD-570/.

## Code availability

Processing scripts used for rodent data are available via Github: https://github.com/henrycope/spaceflight-skin-transcriptomics.

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

## Acknowledgements

H.C. is supported by the Horizon Center for Doctoral Training at the University of Nottingham (UKRI grant no. EP/S023305/1). N.S. was supported by grants from NASA [NSSC22K0250; NSSC22K0278]. A.B. was supported by NASA grant 16-ROSBFP_GL-0005: NNH16ZTT001N-FG Appendix G: Solicitation of Proposals for Flight and Ground Space Biology Research (Award Number: 80NSSC19K0883). C.E.M. thanks the World-Quant Foundation, the GI Research Foundation, NASA (NNX14AH50G, NNX17AB26G, 80NSSC22K0254, NNH18ZTT001N-FG2), the National Institutes of Health (R01MH117406), and the LLS (MCL7001-18, LLS 9238-16). J.K. thanks MOGAM Science Foundation, Boryung, and the National Research Foundation of Korea (RS-2023-00241586).

## Author contributions

Conceptualization: A.B. and H.C.; Methodology: A.B., J.E., H.C., and S.D.; Formal Analysis: A.B., H.C., J.E., S.D., J.T.M., J.C.S., J.P., J.K., S.M.S., and S.R.Z.; Investigation: A.B., C.E.M., H.C., J.E., and S.D.; assisting with writing and literature review: C.W., H.P., H.U., J.C., P.A., and M.C.; astronaut physiological data: S.M.S., S.R.Z., and M.H.; i4 data and omics: C.E.M., C.R.C., E.O., J.W., and J.K.; JAXA data: M.M.; Resource: A.B. and C.E.M.; Original Draft: H.C., J.E., A.B., and N.S.; Review & Editing: H.C., J.E., J.T.M., C.W., H.P., H.U., M.C., J.C., S.R., P.A., S.R.Z., S.M.S., M.H., M.M., C.M., E.O., J.K., C.R.C., J.P., J.C.S., C.E.M., N.J.S., C.R.G.W., A.S., and A.B.; Figures and Visualization: A.B., H.C., J.E., and S.D.; Funding Acquisition: A.B. and C.E.M. (for i4 study); Supervision: A.B. and C.R.G.W.

## Competing interests

J.E. and S.D. are affiliated with Abzu and Abzu is the developer of the QLattice[32], the symbolic regression-method used in this work. All other authors have no competing interests.

## Additional information

¹School of Medicine, University of Nottingham, Derby DE22 3DT, UK. ²Department of Energy Conversion and Storage, Technical University of Denmark, 2800 Kongens Lyngby, Denmark. ³Abzu, Copenhagen 2150, Denmark. ⁴Department of Radiation Medicine, School of Medicine, Georgetown University, Washington D.C., WA 20057, USA. ⁵NASA GeneLab For High Schools Program (GL4HS), Space Biology Program, NASA Ames Research Center, Moffett Field, CA, USA. ⁶Department of Aerospace and Geodesy, TUM School of Engineering and Design, Technical University of Munich, Munich, Germany. ⁷College of Engineering and Haas School of Business, University of California, Berkeley, Berkeley, CA 94720, USA. ⁸School of Engineering and Applied Science, Princeton University, Princeton, NJ 08540, USA. ⁹College of Letters and Science, University of California, Berkeley, Berkeley, CA 94720, USA. ¹⁰Blue Marble Space Institute of Science, Space Biosciences Division, NASA Ames Research Center, Moffett field, CA, USA. ¹¹Space Biosciences Division, NASA Ames Research Center, Moffett field, CA, USA. ¹²Department of Dermatology and Allergy, University Hospital, LMU Munich, 80337 Munich, Germany. ¹³University of Texas Medical Branch, Galveston, TX, USA. ¹⁴Biomedical Research and Environmental Sciences Division, Human Health and Performance Directorate, NASA Johnson Space Center, Houston, TX 77058, USA. ¹⁵IU International University of Applied Sciences, Erfurt and University of Bonn, Bonn, Germany. ¹⁶Transborder Medical Research Center, University of Tsukuba, Ibaraki 305-8575, Japan. ¹⁷Department of Genome Biology, Institute of Medicine, University of Tsukuba, Ibaraki 305-8575, Japan. ¹⁸Department of Physiology, Biophysics and Systems Biology, Weill Cornell Medicine, New York, NY, USA. ¹⁹Laboratory of Virology and Infectious Disease, The Rockefeller University, New York, NY 10065, USA. ²⁰McAllister Heart Institute and Department of Pharmacology, The University of North Carolina at Chapel Hill, Chapel Hill, NC, USA. ²¹Ohio Musculoskeletal and Neurological Institute, Heritage College of Osteopathic Medicine, Ohio University, Athens, OH 45701, USA. ²²School of Chemistry and Biosciences, Faculty of Life Sciences, University of Bradford, Bradford BD7 1DP, UK. ²³St John's Institute of Dermatology, King's College London, Guy's and St Thomas' NHS Foundation Trust, Guy's Hospital, Great Maze Pond, London SE1 9RT, UK. ²⁴Stanley Center for Psychiatric Research, Broad Institute of MIT and Harvard, Cambridge, MA, USA. ²⁵These authors contributed equally: Henry Cope, Jonas Elsborg. ✉e-mail: afshin.beheshti@nasa.gov

