## [Peer Review File · Communications Medicine]

Reviewers' comments:

Reviewer #1 (Remarks to the Author):

The authors performed comprehensive analyses to investigate the effects of spaceflight on skin using public RNA-seq data. They identified 102 key genes involved in spaceflight response and showed that cell cycle- and lipogenesis-associated genes were changed in murine skin. In the analyses of RNA-seq data, they compared the mice to the astronaut. In this study, different tissues (skin, blood, and hair follicles) obtained from different experiments using different mouse strains were compared. This made the interpretation of the results quite complicated and weakened the conclusion validity. As the experimental design, the comparisons are not suitable, while the reviewer understands the limitation of data from samples obtained in the space. This reviewer thinks that correlation of data between murine skin and human skin has not been fully verified. In addition, this study did not reveal whether the identified genes or pathways actually contribute to skin pathogenesis in the space. Also, it should be specified in the Methods section whether the skin samples used in this study contain both epidermis and dermis, or only epidermis.

Some interesting genes were picked up and described in the text,. However, it is hard to find those genes in the comprehensive figures. The authors should reconsider figure layouts.

In Figure 3B, 5A and 6B, many genes have reversed expression patterns depending on the mouse strains (BALB, C3H and C57BL). This indicates that mouse strains in addition to spaceflight influence the gene expressions. To elucidate whether the gene expression changes in each mouse strain are correlated with those in humans, the authors should add figures that show the comparison of the same genes examined in murine skin with those in human skin data from the Inspiration4 studies. Murine skin data should be compared with human skin data.

In Figure 4A and Figure 5B, blood data from JAXA CFE and NASA Twins and hair follicle data from JAXA should not be presented together. Human blood data should be presented relative to mouse blood data. The validity of comparing gene expression in murine skin and human blood should be well discussed.

In Line 362, the authors described that Vitamin D supplementation is already used on the ISS. The authors should clearly mention whether the astronauts who supplied skin samples took Vitamin D supplementation in the space.

Minor comments:

Line 196: Figs. 3C and 3B should be Figs. 3B and 3C.

In line 297-316, citation of "Figure 6" is lacked.

Line 270: FLG should be spelled out at the first appearance in Line 249.

Figure 9: the font size is different in some drug names.

Reviewer #2 (Remarks to the Author):

Review of Cope et al., More than a Feeling: Dermatological Changes Impacted by Spaceflight
The authors describe an analysis of unpublished mouse RNAseq datasets and their comparison with

previously published human datasets. All datasets are RNAseq datasets related to dermatological analyses under space conditions. The heterogeneity of the datasets requires a complex analysis strategy performed by the authors using classical and AI-based analyses. The analysis strategy is plausible, and the presentation of results is largely appropriate. However, the presentation of the results lacks precision, which is absolutely necessary due to the complex analysis situation.

1. The missing description of important abbreviations makes it difficult to follow the article without bioinformatics or immunological knowledge. Abbreviations are not explained in the text: Gene Ontology Biological Process (GOBP), differential expressed genes (DEG), Peripheral blood mononuclear cells (PBMC), log fold change (LFC; are those log₂ or log₁₀?), Molecular Signatures Database (MSigDB) etc.

2. Abbreviations that are explained are not explained at their first occurrence (e.g. FDR).

3. In addition, facts are presented in an incomprehensibly abbreviated manner: “union of intra-MHU-2 combinations”, in figure 2

4. Different FDR thresholds are applied. Here, the use of FDRs above the 10% threshold is questionable and requires at least a detailed explanation: Figure 5, FDR ≤ 0.25

Lines 126-130: Please indicate the cluster method and genes (genes in text or in a supplement).

Lines 138-139 : Please give the GLYCAM1 P-value.

Line 166 : Please report the results for the 102 key genes in a supplement.

Lines 175-176 “For example, in the case of the gene with the lowest p-value, LAMA1 (Laminin Subunit Alpha 1)”: Please indicate the significance level or P-value and the compared conditions.

Line 166: Please mention string and the confidence level (0.7).

Lines 204-205 “It is interesting to note the significantly lower upregulation of these metabolic pathway genes for female mice.”: Please indicate the significance level or P-value

Lines 231-294: The data interpretation is unclear. The total number of regulated genes in the studies is needed for the interpretation of the overlap.

Lines 330-331: “One of these models demonstrated how the upregulation of D2HGDH” the bioinformatical context is needed: number and size of models.

Lines 356-357: “Calcitriol and L-asparaginase exhibited significant activation scores for the majority of the datasets”: please give the scores at least in a supplement.

Lines 644: 0.05 is a normal threshold for FDR and not “highly significant”.

Lines 665-670: Please indicate if the analysis is done by your team or is done in the previous study.

Figures 1 B and 3 D: text size and colour selection is suboptimal.

Dear Editor and Reviewers,

We thank both the editor and reviewers for the comments. We believe based on these final revisions we have made that this manuscript is now stronger and much improved. We have addressed the comments by the reviewers and our responses appear in red font below the original reviewer comment. We look forward to the next steps.

Afshin Beheshti, PhD

Referee expertise:

Referee #1: Spaceflight, genomics, cell biology, skin biology

Referee #2: Spaceflight, genomics, cell biology, skin biology

Reviewers' comments:

Reviewer #1 (Remarks to the Author):

The authors performed comprehensive analyses to investigate the effects of spaceflight on skin using public RNA-seq data. They identified 102 key genes involved in spaceflight response and showed that cell cycle- and lipogenesis-associated genes were changed in murine skin. In the analyses of RNA-seq data, they compared the mice to the astronaut. In this study, different tissues (skin, blood, and hair follicles) obtained from different experiments using different mouse strains were compared. This made the interpretation of the results quite complicated and weakened the conclusion validity. As the experimental design, the comparisons are not suitable, while the reviewer understands the limitation of data from samples obtained in the space. This reviewer thinks that correlation of data between murine skin and human skin has not been fully verified.

We thank the reviewer for their comment. We have substantially revised the manuscript based on your comments and provided the analysis with comparison that make more sense. We have also revised and refined our analysis further to address your concerns.

1. In addition, this study did not reveal whether the identified genes or pathways actually contribute to skin pathogenesis in the space.

Thank you for the reviewer's comment, we have modified language to frame the contributions of the investigation more thoroughly, including in the abstract. The potential links to dermatological manifestations in astronauts have been made clearer, especially in the discussion.

1. Also, it should be specified in the Methods section whether the skin samples used in this study contain both epidermis and dermis, or only epidermis.

We thank the reviewer, we made use of public data, and unfortunately we do not have detailed accounts of the dissection procedures beyond the metadata in the database from which the data was obtained (which we point the reader to in the methods section). A note has also been added to the study limitations.

2. Some interesting genes were picked up and described in the text,. However, it is hard to find those genes in the comprehensive figures. The authors should reconsider figure layouts.

At the request of the reviewer we have revised figure layouts and visibility is now improved. For example, Figure 3A now shows only cross-mission genes mentioned throughout the text, and some heat maps have been moved to supplemental so that larger text can be shown for gene names in the main figures.

3. In Figure 3B, 5A and 6B, many genes have reversed expression patterns depending on the mouse strains (BALB, C3H and C57BL). This indicates that mouse strains in addition to spaceflight influence the gene expressions. To elucidate whether the gene expression changes in each mouse strain are correlated with those in humans, the authors should add figures that show the comparison of the same genes examined in murine skin with those in human skin data from the Inspiration4 studies. Murine skin data should be compared with human skin data.

Thanks to the excellent reviewer suggestion in Figure 5 we now show the skin health genes in human data from the Inspiration4 mission alongside the murine skin data.

4. In Figure 4A and Figure 5B, blood data from JAXA CFE and NASA Twins and hair follicle data from JAXA should not be presented together. Human blood data should be presented relative to mouse blood data. The validity of comparing gene expression in murine skin and human blood should be well discussed.

Based on the comments of the reviewer and studies finding differences between gene expression in skin and blood samples (e.g. PMIDs: 29228364, 26091259, 22299064), we have prioritized demonstrating key genes involved in space flight response in murine skin and subsequently investigated the modifications in expression of these genes both in murine and human skin and hair follicles; but given the availability of blood samples from astronauts, we have also included the latter to our analysis. Moreover, time points and experimental conditions/study design for analysis of spaceflight response of human skin and murine skin

differed. Due to this limitation, although we did try to correlate the results between murine skin and human skin, and seek common alterations in gene expression, a one on one comparison was not the main focus of this study. We have rather prioritized on outlining the results obtained under different experimental conditions, side by side. And, in cases where we have seen a global signature in skin tissue, such as a specific gene down regulation in both human and murine skin, we did try to explain some of the alterations in murine skin with the results acquired from human blood, assuming that a similar change would be observed in mice blood. Nevertheless, despite the differences in study design, we did still seek global signatures in all datasets; human blood & skin and mice skin. In addition to refocusing the figures on skin and hair follicle data by moving blood sample data into supplemental figures, we have also rephrased the language so that we are not implying direct comparison between blood and skin.

5. In Line 362, the authors described that Vitamin D supplementation is already used on the ISS. The authors should clearly mention whether the astronauts who supplied skin samples took Vitamin D supplementation in the space.

We have added the following text to the that section to provide more clarity for vitamin D supplementation: In the blood markers section of the results we write “Notably, this decrease in vitamin D is observed despite consistent programmatic vitamin D supplementation since the collection of blood samples⁶².” and in the i4 section of the methods we write “Records of vitamin D supplement consumption were not available for the i4 crew.”

6. Minor comments:
 - a. Line 196: Figs. 3C and 3B should be Figs. 3B and 3C
 - b. In line 297-316, citation of "Figure 6" is lacked
 - c. Line 270: FLG should be spelled out at the first appearance in Line 249
 - d. Figure 9: the font size is different in some drug names.

We have addressed these minor comments and thank the reviewer for their thoroughness.

Reviewer #2 (Remarks to the Author):

Review of Cope et al., More than a Feeling: Dermatological Changes Impacted by Spaceflight
The authors describe an analysis of unpublished mouse RNAseq datasets and their comparison with previously published human datasets. All datasets are RNAseq datasets related to dermatological analyses under space conditions. The heterogeneity of the datasets requires a complex analysis strategy performed by the authors using classical and AI-based analyses. The analysis strategy is plausible, and the presentation of results is largely appropriate. However, the presentation of the results lacks precision, which is absolutely necessary due to the complex analysis situation.

We thank the reviewer for their insightful comments. We have substantially revised the manuscript to improve the presentation to bring more precision with the results. We have also refined our analysis to further make the presentation and the results more clear and precise.

1. The missing description of important abbreviations makes it difficult to follow the article without bioinformatics or immunological knowledge. Abbreviations are not explained in the text: Gene Ontology Biological Process (GOBP), differential expressed genes (DEG), Peripheral blood mononuclear cells (PBMC), log fold change (LFC; are those \log_2 or \log_{10} ?), Molecular Signatures Database (MSigDB) etc.

We thank the reviewer, and have added acronyms throughout, such as for: GOBP, DEGs, PBMCs, LFC, FDR, MSigDB, ISS, and JAXA.

2. Abbreviations that are explained are not explained at their first occurrence (e.g. FDR).

In addition to introducing additional acronyms, we have edited to explain them during their first occurrences.

3. In addition, facts are presented in an incomprehensibly abbreviated manner: "union of intra-MHU-2 combinations", in figure 2 4. Different FDR thresholds are applied. Here, the use of FDRs above the 10% threshold is questionable and requires at least a detailed explanation: Figure 5, FDR ? 0.25

To accommodate the reviewer's comment, we have updated the plots to show $FDR < 0.05$ and have provided all the Gene Set Enrichment Analysis (GSEA) values as supplemental data for readers to reference. Throughout the paper we use $FDR < 0.1$ for defining significant genes, and $FDR < 0.05$ for defining significant pathways. Additionally, we removed the term "union of intra-MHU-2 combinations" as it was indeed confusing; we have tried to simplify processes throughout, including the derivation of "key genes", which are now referred to as "cross-mission genes", which we believe to be easier to interpret.

While we have decided on a more stringent cutoff of $FDR < 0.05$ for pathways, we note that the scientists that created the GSEA algorithm do consider an appropriate cutoff for statistics to be $FDR < 0.25$.

The GSEA FAQ question that specifically addresses this point can be found here: https://software.broadinstitute.org/cancer/software/gsea/wiki/index.php/FAQ#Why_does_GSEA_use_a_false_discovery_rate_.28FDR.29_of_0.25_rather_than_the_more_classic_0.05.3F. And there is further information here: https://www.gsea-msigdb.org/gsea/doc/GSEAUUserGuideFrame.html?Interpreting_GSEA.

4. Lines 126-130: Please indicate the cluster method and genes (genes in text or in a supplement).

We thank the reviewer for their comment. All gene statistics are now provided as supplemental, and the clustering method is discussed within the methods section.

5. Lines 138-139 : Please give the GLYCAM1 P-value.

The adjusted p-value has now been provided in the text.

6. Line 166 : Please report the results for the 102 key genes in a supplement.

Key genes (now “cross-mission genes”) information is now provided within the supplemental results file.

7. Lines 175-176 "For example, in the case of the gene with the lowest p-value LAMA1 (Laminin Subunit Alpha 1)": Please indicate the significance level or P-value and the compared conditions.

In the revised version, we do not highlight the significance of LAMA1 due to other genes from the cross-mission list having more pronounced effects on the prediction of spaceflight status as well as having more clear interpretations. We highlight these using the QLattice models and the corresponding AUC scores.

8. Line 166: Please mention string and the confidence level (0.7).

We felt that the network method was overly complex without adding much to the biological interpretation, so in line with reviewer comments on the complexity of the paper, we have since removed the use of StringDB.

9. Lines 204-205 "It is interesting to note the significantly lower upregulation of these metabolic pathway genes for female mice.": Please indicate the significance level or P-value Lines 231-294: The data interpretation is unclear. The total number of regulated genes in the studies is needed for the interpretation of the overlap.

We have dropped the word “significantly lower” and rephrased to “weaker” upregulation. Figure 2A provides details on quantities of significantly regulated genes within datasets.

10. Lines 330-331: "One of these models demonstrated how the upregulation of D2HGDH" the bioinformatical context is needed: number and size of models.

We thank the reviewer for their comment, we have added some text to express that this was within the top 10 2-gene models, as well as explicitly stating the corresponding AUC score.

11. Lines 356-357: "Calcitriol and L-asparaginase exhibited significant activation scores for the majority of the datasets": please give the scores at least in a supplement.

Activation scores and other statistics from the IPA analysis are now provided as a supplemental file.

12. Lines 644: 0.05 is a normal threshold for FDR and not "highly significant".

This is a fair criticism, the word "highly" has now been removed throughout in the context of significance.

13. Lines 665-670: Please indicate if the analysis is done by your team or is done in the previous study.

We thank the author for their suggestion, we now state that the data processing was reported in a previous paper, and we include a citation.

14. Figures 1 B and 3 D: text size and colour selection is suboptimal.

While 3D is no longer included as part of the simplification of displayed results, the colors have been adjusted in 1B to improve readability.

REVIEWERS' COMMENTS:

Reviewer #1 (Remarks to the Author):

In the revised manuscript, the authors have largely responded to the reviewers' comments. There are minor points that the authors should revise.

1. In Figure 1A, the text and graphic elements are overlapping.
2. In Figures 5A and 5B, please improve the arrangement and size of the schematics depicting the astronaut, skin, and hair.
3. In Figure 4A, the authors have included many gene expression results. However, due to the excessive amount, they are scarcely mentioned in the text. Please refine the selection of genes displayed in the figure to make it easier to read.

Reviewer #2 (Remarks to the Author):

All my points of criticism have been implemented. I therefore recommend the article for publication.

Dear Editor and Reviewers,

We thank both the editor and reviewers for the comments. We believe based on these final revisions we have made that this manuscript is now stronger and much improved. We have addressed the comments by the reviewers and our responses appear in red font below the original reviewer comment. We look forward to the next steps.

Afshin Beheshti, PhD

REVIEWERS' COMMENTS:

Reviewer #1 (Remarks to the Author):

In the revised manuscript, the authors have largely responded to the reviewers' comments. There are minor points that the authors should revise.

We thank the reviewer for their comments which has greatly improved the manuscript.

1. In Figure 1A, the text and graphic elements are overlapping.

We have made the update in Figure 1A to avoid the overlap.

2. In Figures 5A and 5B, please improve the arrangement and size of the schematics depicting the astronaut, skin, and hair.

We have made rearranged and improved of the schematics as the reviewer has request.

3. In Figure 4A, the authors have included many gene expression results. However, due to the excessive amount, they are scarcely mentioned in the text. Please refine the selection of genes displayed in the figure to make it easier to read.

We have refined this figure to only display the selected genes.

Reviewer #2 (Remarks to the Author):

All my points of criticism have been implemented. I therefore recommend the article for publication.

We thank the reviewer for their comments which has greatly improved the manuscript.